# NEK2 inhibition triggers anti-pancreatic cancer immunity by targeting PD-L1

Xiaozhen Zhang[1,2,3,4,5,6,7], Xing Huang [1,2,3,4,5,6,7✉], Jian Xu[1,2,3,4,5,6], Enliang Li[1,2,3,4,5,6], Mengyi Lao[1,2,3,4,5,6], Tianyu Tang[1,2,3,4,5,6], Gang Zhang[1,2,3,4,5,6], Chengxiang Guo[1,2,3,4,5,6], Xiaoyu Zhang[1,2,3,4,5,6], Wen Chen[1,2,3,4,5,6], Dipesh Kumar Yadav [1,2,3,4,5,6], Xueli Bai [1,2,3,4,5,6✉] & Tingbo Liang[1,2,3,4,5,6✉]

Despite the substantial impact of post-translational modifications on programmed cell death 1 ligand 1 (PD-L1), its importance in therapeutic resistance in pancreatic cancer remains poorly defined. Here, we demonstrate that never in mitosis gene A-related kinase 2 (NEK2) phosphorylates PD-L1 to maintain its stability, causing PD-L1-targeted pancreatic cancer immunotherapy to have poor efficacy. We identify NEK2 as a prognostic factor in immunologically "hot" pancreatic cancer, involved in the onset and development of pancreatic tumors in an immune-dependent manner. NEK2 deficiency results in the suppression of PD-L1 expression and enhancement of lymphocyte infiltration. A NEK binding motif (F/LXXS/T) is identified in the glycosylation-rich region of PD-L1. NEK2 interacts with PD-L1, phosphorylating the T194/T210 residues and preventing ubiquitin-proteasome pathway-mediated degradation of PD-L1 in ER lumen. NEK2 inhibition thereby sensitizes PD-L1 blockade, synergically enhancing the anti-pancreatic cancer immune response. Together, the present study proposes a promising strategy for improving the effectiveness of pancreatic cancer immunotherapy.

[1] Department of Hepatobiliary and Pancreatic Surgery, the First Affiliated Hospital, School of Medicine, Zhejiang University, Hangzhou, Zhejiang, China. [2] Zhejiang Provincial Key Laboratory of Pancreatic Disease, School of Medicine, Zhejiang University, Hangzhou, Zhejiang, China. [3] Innovation Center for the Study of Pancreatic Diseases, Hangzhou, Zhejiang, China. [4] Zhejiang Clinical Research Center of Hepatobiliary and Pancreatic Disease, Hangzhou, Zhejiang, China. [5] Zhejiang University Cancer Center, Hangzhou, Zhejiang, China. [6] Research Center for Healthcare Data Science, Zhejiang Lab, Hangzhou, Zhejiang, China. [7] These authors contributed equally: Xiaozhen Zhang, Xing Huang. ✉email: huangxing66@zju.edu.cn; shirleybai@zju.edu.cn; liangtingbo@zju.edu.cn

Pancreatic ductal adenocarcinoma (PDAC) is the most common exocrine cell tumor of the pancreas and the 7th leading cause of death in cancer patients worldwide[1,2]. Despite advances in diagnosis, perioperative management, radiotherapy techniques, and systematic treatments that enhance the survival rate of pancreatic cancer, the numbers of such cases and related deaths have steadily increased[1,3]. Immunotherapy is considered the most promising strategy to reverse this trend, especially immune checkpoint blockade targeting programmed cell death 1 (PD-1) and its ligand 1 (PD-L1, also known as B7-H1)[4,5]. Multiple clinical trials of PD-1/PD-L1 blockade have been launched as treatments for melanoma, lung cancer, and kidney cancer, with promising clinical results[5–7]. However, PD-1/PD-L1 blockade has so far demonstrated limited effectiveness in PDAC. For example, Royal et al. reported that no in-group patients with advanced pancreatic cancer responded, preventing an evaluation of the effectiveness of anti-PD-L1 antibodies[8]. Furthermore, a phase II randomized clinical trial of 65 patients with metastatic PDAC suggested that the objective response rate of patients receiving durvalumab monotherapy was 0%, although that of patients receiving combination therapy was 3.1%[9]. Thus, an in-depth understanding of the detailed mechanism of PD-L1-based combinatorial therapy may contribute to the development of novel therapeutic strategies[10].

Accumulating evidence has demonstrated that the therapeutic efficacy of drugs targeting PD-L1 can be seriously affected by post-translational modifications (PTMs), including ubiquitination, phosphorylation, and glycosylation. Such PTMs play critical roles in regulating the stability, internalization, and localization of PD-L1, in addition to its physiological and pathological functions[11]. For instance, N-glycosylation of PD-L1 facilitates its folding, intracellular transport, and function[12–14]. Epidermal growth factor (EGF) signaling upregulates b-1,3-N-acetylglucosaminyltransferase (B3GNT3) expression that promotes N-glycosylation at positions N192, N200, and N219 of PD-L1 in triple-negative breast cancer cells. This stabilizes PD-L1 through the prevention of glycogen synthase kinase 3 beta (GSK3β) binding and phosphorylation of PD-L1 at the T180 and S184 residues, subsequently recognized by beta-transducin repeat containing E3 ubiquitin-protein ligase (BTRCP)-mediated PD-L1 poly-ubiquitination and degradation[15,16]. More importantly, the combination of EGFR inhibition and PD-L1 blockade significantly enhances mono-drug therapeutic efficacy. In addition, metformin-activated AMP-activated protein kinase directly phosphorylates PD-L1 at the S195 residue, inducing abnormal PD-L1 glycosylation that leads to PD-L1 degradation through an endoplasmic reticulum-associated protein degradation pathway[16]. Therefore, identification of critical regulators of PD-L1 PTM in pancreatic cancer is urgently required to improve the efficacy of PD-L1 blockade in this specific cancer type.

Never in mitosis gene A (NIMA)-related kinase 2 (NEK2) is a member of NEKs, a serine-threonine kinase family including proteins structurally related to the essential mitotic regulator[17]. NEK2 is a multifunctional protein physiologically involved in cell cycle regulation, such as centrosome duplication and separation[17,18], microtubule stabilization[19], kinetochore attachment[20], in addition to spindle assembly checkpoint[21,22]. NEK2 is also considered a key factor in the maintenance of normal development and function of B cells[23]. Intriguingly, aberrant expression of NEK2 and NEK2-mediated phosphorylation of downstream proteins (e.g. p53 at Ser315[24] and GAS2L1 at Ser352[25]) are frequently observed to be associated with tumor initiation, progression, metastasis, and consequently poor prognosis in multiple cancers, including but not limited to, breast cancer[26,27], colorectal cancer[28], and PDAC[29]. However, the potential role of NEK2 in cancer immune resistance so far remains undefined.

In this work, we demonstrate that PD-L1 is a substrate of NEK2, and combinatorial inhibition of NEK2 and PD-L1 largely improves the therapeutic efficacy of pancreatic cancer in pre-clinical models.

## Results

**NEK2 is a prognostic factor in pancreatic cancer**. To assess whether members of the NEK family (from *NEK1* to *NEK11*) are associated with pancreatic cancer prognosis, we first compared their expression levels in tumors and normal tissue using pancreatic adenocarcinoma (PAAD) datasets from The Cancer Genome Atlas (TCGA) database. Bioinformatic analyses across multiple cancers demonstrated that NEK2 is the most significantly upregulated member of the NEK family in PAAD compared with expression in normal tissues (Fig. 1a and Supplementary Fig. 1a). The elevated expression of NEK2 was further validated in paired PDAC tissue samples by immunohistochemistry (Fig. 1b, c). More importantly, the significantly negative prognostic relevance of NEK2 was observed in a tumor tissue microarray (Fig. 1d). The analysis of TCGA data further confirmed that the higher expression of NEK2 was correlated with inferior overall survival (OS) (Fig. 1e and Supplementary Fig. 1b). Furthermore, high expression of NEK2 was also found to be significantly associated with vascular invasion and tumor grade as opposed to TNM stage and other clinical indices in multiple patient cohorts of pancreatic cancer (Table 1 and Supplementary Fig. 2a–c). Together, these results suggest that the elevated expression of NEK2 is unfavorable for patients with PDAC.

**NEK2 expression indicates poor prognosis in immunologically "hot" pancreatic cancer**. Since tumors are classified as immunologically "cold" (CD8[+] T cell-decreased) or "hot" (CD8[+] T cell-enriched), and their immune status is closely associated with tumor prognosis[30], we further investigated the prognostic role of NEK2 in increased or decreased CD8[+] T cell pancreatic cancer. Intriguingly, NEK2 expression was only significantly associated with OS in immunologically "hot" cells, rather than those that were "cold" (Fig. 1f, g). Furthermore, the opposite relationship was observed between the expression of NEK2 and the number of immune effector cells (Fig. 1h). Interestingly, NEK2 was also expressed in multiple immunosuppressive populations, such as myeloid-derived suppressor cells (MDSCs), dendritic cells (DCs), and macrophages, although the expression-positive rate was lower than that in tumor cells (Supplementary Fig. 3a, b). Furthermore, GO-enrichment analysis revealed that NEK2 overexpression led to a dramatic increase in the expression of proteins related to the positive regulation of leukocyte proliferation, migration, and activation, and lymphocyte activation (Supplementary Fig. 3c, d). These results indicate that NEK2 may influence the efficacy of immunotherapy in multiple ways. Additionally, given the critical role of tumor mutation burden (TMB) in the immunotherapy response, we next investigated the prognostic relevance of NEK2 in immunologically "hot" pancreatic cancer with high or low TMB, respectively. Surprisingly, NEK2 displayed a significant correlation with poor prognosis in low mutational tumors as opposed to high mutational tumors, further indicating the precise prognostic value of NEK2 in specific patients (Supplementary Fig. 4a, b). Taken together, NEK2 was found to be a prognostic factor in immunologically "hot" pancreatic cancer.

**NEK2 deficiency improves pancreatic cancer immunogenicity**. A KPC-NEK2 knockdown (KD) pancreatic cancer cell line, stably transfected with *Nek2* double nickase plasmids, was generated to determine whether NEK2 is associated with an anti-tumor immune response. In comparison with wild-type (WT) control

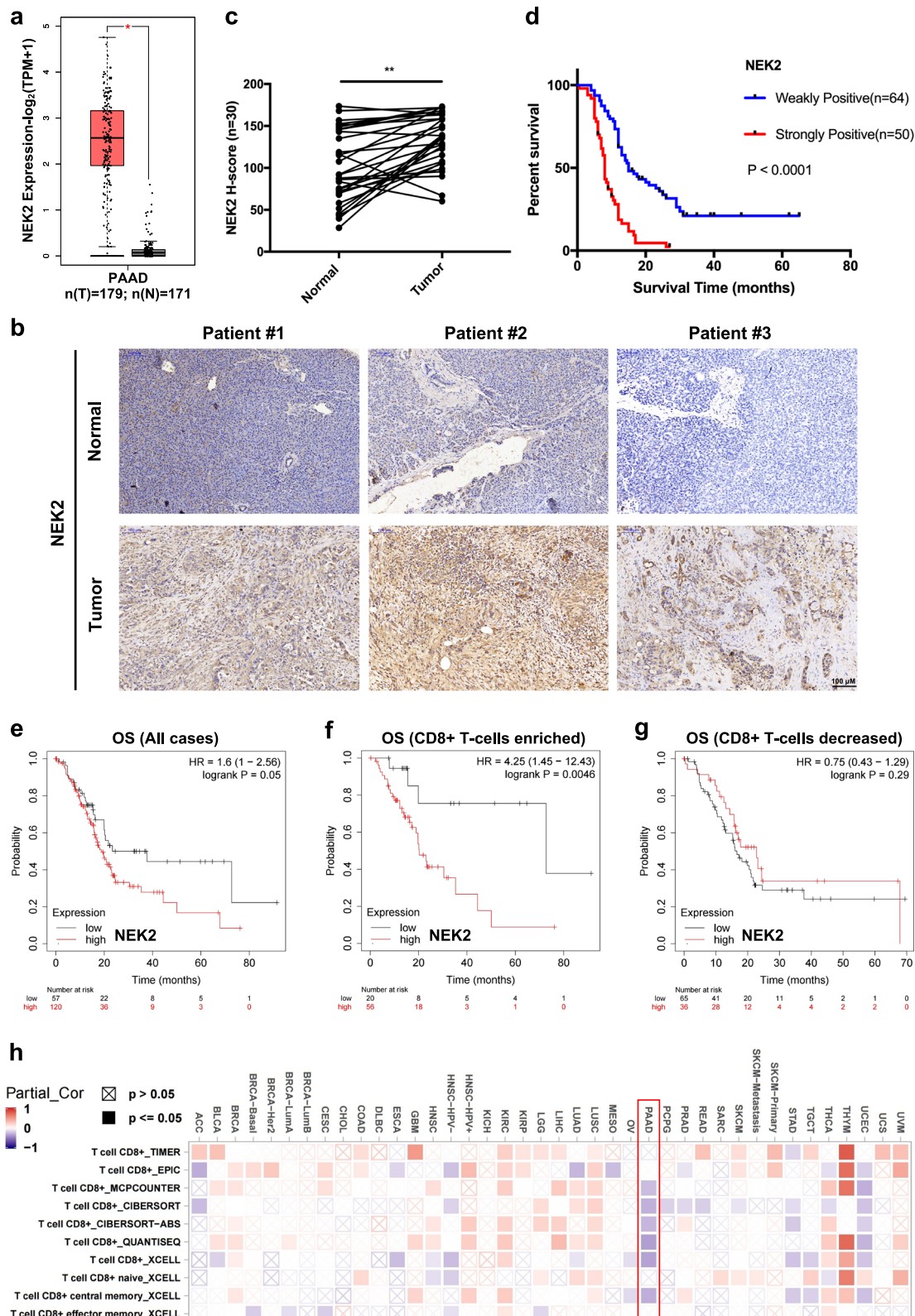

cells, NEK2 KD-cells displayed significantly weakened resistance to T cell-mediated tumor cell-destruction in vitro (Supplementary Fig. 5a, b). The growth of NEK2 KD cell-derived tumors was consistently significantly repressed in immunocompetent C57BL/6 mice but not in immunodeficient nude mice (Fig. 2a–h). Flow cytometry further revealed a significant increase in the number and function of CD8$^+$ T cells in NEK2 KD cell-derived tumors

(Fig. 2i, j). Furthermore, immunocompetent mice with such tumors exhibited markedly extended survival that was not observed in immunodeficient nude mice (Fig. 2k–m). Taken together, the results suggest that NEK2 is involved in the development and progression of PDAC in an immune-dependent manner, with deletion of NEK2 improving pancreatic cancer immunogenicity.

**Fig. 1 NEK2 is a prognostic factor for immunologically "hot" pancreatic cancer. a–c** Expression profile of NEK2 in pancreatic cancer. Relative *NEK2* expression in pancreatic cancer was analyzed using large-scale RNA-Seq datasets of PDAC from the TCGA database (n (T) = 179, n (N) = 171; n: number of patients) (**a**). NEK2 expression was measured in paired tumor and normal pancreatic tissues by IHC staining, with representative images (**b**) and statistical results (**c**) shown (n = 30; n: number of patients) (N: Normal pancreatic tissue; T: Pancreatic tumor tissue). Scale bars: 100 µm. **d** Correlation of NEK2 with prognostic factors in pancreatic cancer. Tissue microarray analysis of the prognostic role of NEK2 in pancreatic cancer (n = 64 weakly positive; n = 50 strongly positive) (p < 0.0001). **e** Overall survival (OS) of patients with pancreatic cancer with high or low expression of NEK2 (n = 177). **f** Overall survival (OS) of pancreatic cancer patients with high numbers of CD8$^+$ T cells and with high or low expression of NEK2 (n = 76). **g** Overall survival (OS) of pancreatic cancer patients with decreased CD8$^+$ T cells and with high or low expression of NEK2 (n = 101). **h** Bioinformatics analysis of the correlation between NEK2 and immune effector cells using TCGA datasets. In **a**, data are represented as boxplots where the middle line is the median; the lower and upper hinges correspond to the first and third quartiles; the upper whisker extends from the hinge to the largest value no further than 1.5 × IQR from the hinge (where IQR is the inter-quartile range); the lower whisker extends from the hinge to the smallest value at most 1.5 × IQR of the hinge, while data beyond the end of the whiskers are outlying points that are plotted individually. *P < 0.05, **P < 0.01, ***P < 0.001 using a two-tailed t-test; ns: not significant. Kaplan–Meier method and a Gehan–Breslow–Wilcoxon test are indicated in **d**. The Hazard Ratios (HR) and p-values by the log-rank (Mantel–Cox) test are indicated in **e–g**.

**Table 1 Clinicopathological relevance of NEK2 in PDAC patients.**

| Variable | NEK2 | | |
|---|---|---|---|
| | Low (*H*-score ≤ 130) | High (*H*-score > 130) | *P* value |
| *Gender* | | | 0.925 |
| Male, n (%) | 44(57.9) | 44(57.1) | |
| Female, n (%) | 32(42.1) | 33(42.9) | |
| *Age, years* | | | 0.461 |
| >60, n (%) | 64(84.2) | 68(88.3) | |
| ≤60, n (%) | 12(15.8) | 9(11.7) | |
| *BMI, kg/m²* | | | 0.238 |
| <18.5, n (%) | 16(22.2) | 9(12.7) | |
| 18.5–23.9, n (%) | 35(48.6) | 43(60.6) | |
| >23.9, n (%) | 21(29.2) | 19(26.8) | |
| *TNM stage* | | | 0.239 |
| I, n (%) | 14(18.4) | 11(14.5) | |
| II, n (%) | 40(52.6) | 48(63.2) | |
| III, n (%) | 19(25.0) | 11(14.5) | |
| IV, n (%) | 3(4.0) | 6(7.9) | |
| *Vascular invasion* | | | **0.043** |
| Yes, n (%) | 30(39.0) | 42(55.3) | |
| No, n (%) | 47(61.0) | 34(44.7) | |
| *Nerve invasion* | | | 0.086 |
| Yes, n (%) | 48(63.2) | 57(76.0) | |
| No, n (%) | 28(36.8) | 18(24.0) | |
| *Serum CA12-5, U/mL* | | | 0.427 |
| ≥35, n (%) | 19(29.2) | 17(23.3) | |
| <35, n (%) | 46(70.8) | 56(76.7) | |
| *Serum CA19-9, U/mL* | | | 0.907 |
| ≥37, n (%) | 63(81.8) | 60(81.1) | |
| <37, n (%) | 14(18.2) | 14(18.9) | |
| *Serum CEA, U/mL* | | | 0.714 |
| ≥5, n (%) | 24(35.8) | 24(32.9) | |
| <5, n (%) | 43(64.2) | 49(67.1) | |
| *Tumor differentiation* | | | **0.045** |
| Well, n (%) | 3(5.6) | 5(6.8) | |
| Moderate, n (%) | 53(72.6) | 39(52.7) | |
| Poor, n (%) | 17(23.3) | 30(40.5) | |
| *Recurrence* | | | 0.982 |
| Yes, n (%) | 42(72.4) | 39(72.2) | |
| No, n (%) | 16(27.6) | 15(27.8) | |

One-way and repeated-measures analysis of variance (ANOVA) was performed in Table 1. The bold values indicate p < 0.05.

**NEK2 inhibition activates an anti-pancreatic cancer immune response.** To investigate the influence of NEK2 inhibition on pancreatic tumorigenesis and growth, pancreatic cancer cells both with and without pretreatment using a NEK2 inhibitor were injected s.c. into C57BL/6 and nude mice, respectively (Fig. 3a). Pretreatment with NEK2 inhibitor resulted in a lower incidence of tumors and postponed tumor occurrence (Fig. 3b), whereas such a trend was not observed in nude mice (Fig. 3c, d). Furthermore, tumor-bearing C57BL/6 and nude mice were individually administered with NEK2 inhibitor and tumor growth was significantly suppressed in C57BL/6 mice (Fig. 3e–g) but not in nude mice (Fig. 3h–j). Moreover, the NEK2 inhibitor imposed no significant effect on the weight of mice compared with the control group (Fig. 3k, l). Additional immunohistochemical assays indicated a significant decrease in PD-L1 expression and increased CD8$^+$ T cell infiltration in NEK2 inhibitor-treated tumors, suggesting that NEK2 inhibition may result in downregulation of PD-L1 and an enhancement of T cell function (Fig. 3m–o). In support of this, the specific NEK2 inhibitor significantly enhanced T cell-mediated tumor cell-destruction in vitro (Supplementary Fig. 5c, d). Collectively, the results demonstrate that NEK2 inhibition suppressed the onset and development of pancreatic cancer, possibly associated with the decreased expression of PD-L1.

**NEK2 positively correlates and interacts with PD-L1 in pancreatic cancer.** Since PD-L1 is a well-recognized and quite important factor in the mediation of tumor immune resistance, we further investigated the regulatory effects of NEK2 on PD-L1 expression. The relationship between NEK2 and PD-L1 was first identified as co-occurrence at a genomic alteration level by mutual exclusivity analysis (Supplementary Fig. 6a). It was further confirmed to have a positive association in multiple assays, including, but not limited to, at the transcriptional level in the TCGA database (Supplementary Fig. 6b) and at the protein level in paired clinical tissue samples (Fig. 4a), in addition to tumor tissue microarrays (Fig. 4b–d). Moreover, a significant correlation was observed in KPC (Kras$^{LSL-G12D}$; Trp53$^{LSL-R172H}$; Pdx1-Cre) and KTC (Kras$^{LSL-G12D/+}$; Tgfbr2$^{flox/wt}$; Ptf1a-Cre) autochthonous pancreatic tumors in a genetically engineered mouse model (GEMM) (Supplementary Fig. 6c–e). Expression levels of PD-L1 were significantly reduced in *NEK2* KD tumors compared with WT control tumors in a xenograft mouse model (Supplementary Fig. 7a, b). Furthermore, an endogenous interaction between NEK2 and PD-L1 in multiple pancreatic cell lines was observed (Fig. 4f, g). Moreover, glutathione S-transferase (GST)-pull down assay showed that NEK2 bound to PD-L1 directly (Fig. 4h). Consistently, Duo-link assay demonstrated the real binding between NEK2 and PD-L1 in the cell (Fig. 4i). In further investigation of the localization of NEK2–PD-L1 interaction, no signals were detected with antibodies against both cytosolic and luminal proteins after trypsinization in the permeable fraction; instead, signals of the cytosolic domain of IRE1α were rapidly

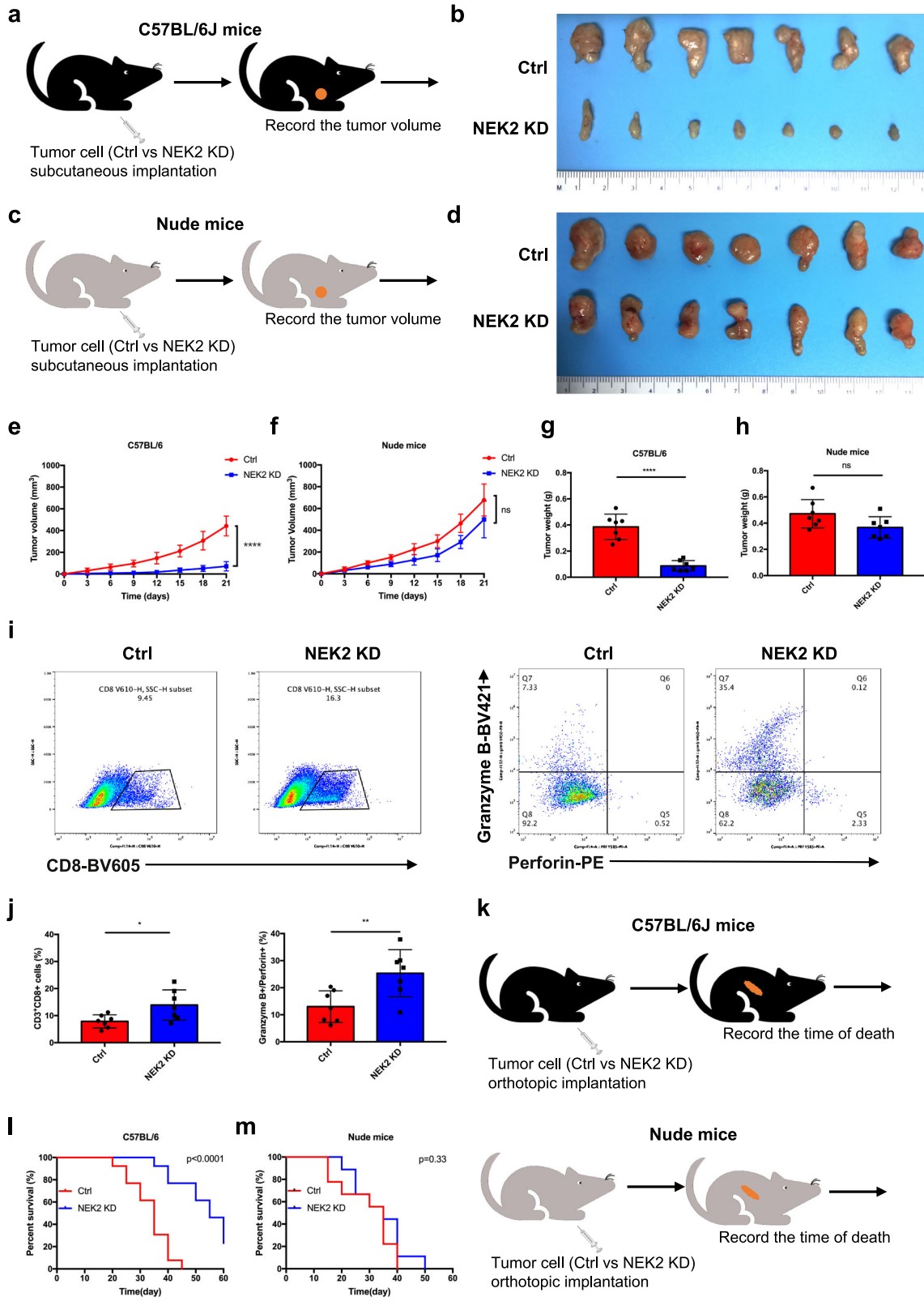

reduced after trypsinization, whereas signals of NEK2 and HSP90B1 were maintained in the non-permeable fraction (Supplementary Fig. 8a). Taken together, the results indicate a positive correlation and direct interaction of NEK2 with PD-L1 inside the ER lumen in pancreatic cancer.

**NEK2 inhibits ubiquitination-mediated proteasomal degradation of PD-L1.** To further investigate the manner in which NEK2 regulates PD-L1, we measured the expression levels of PD-L1 in NEK2 inhibitor-treated, *NEK2* KD, or overexpressed cells. NEK2 inhibition or depletion decreased PD-L1 expression in pancreatic

**Fig. 2 NEK2 deficiency improves pancreatic cancer immunogenicity. a, c** Schematic protocols displaying WT and NEK2-depleted pancreatic cancer cells separately, and s.c. injection into immunocompetent and immunodeficient mice ($n = 7$). **b, e** Representative images of tumors and growth curves of immunocompetent mice. Tumors were measured at specified time points then dissected at the endpoint ($n = 7$). **d, f** Representative images of tumors and growth curves of tumors from immunodeficient mice. Tumors were measured at specified time points and dissected at the endpoint ($n = 7$). **g, h** The weight of tumors from immunocompetent and immunodeficient mice was reported at the endpoint ($n = 7$). **i, j** Representative images and statistical results of tumor-infiltrating lymphocytes ($n = 7$). **k** Schematic protocols showing WT and NEK2-depleted pancreatic cancer cells separately injected into immunocompetent and immunodeficient mice ($n = 13$). **l, m** Survival of immunocompetent and immunodeficient mice bearing NEK2-depleted pancreatic cancer cells ($n = 13$). Kaplan–Meier survival curves with log-rank test assessing the significance between WT and NEK2 KD immunocompetent (**l**) ($p < 0.0001$) and immunodeficient (**m**) ($p = 0.33$) mice. Results represent means ± SD of one representative experiment in **e–j**. All data are representative of three independently performed experiments. *$P < 0.05$, **$P < 0.01$, ***$P < 0.001$ using a two-tailed $t$-test; ns: not significant. Kaplan–Meier method and a Gehan–Breslow–Wilcoxon test are indicated in **m** and **l**.

cancer cells (Fig. 5a, b), while the overexpression of NEK2 induced the upregulation of PD-L1 (Fig. 5c). Interestingly, downregulated PD-L1 expression caused by NEK2 depletion or inhibition could be restored by treatment with the proteasome inhibitor MG132, suggesting that NEK2 prevents proteasome-mediated degradation of PD-L1 (Fig. 5d, e). Indeed, PD-L1 in both *NEK2* KD and inhibition groups exhibited a shorter half-life than control groups, a reduction that was also rescued by MG132 (Fig. 5f–i). Furthermore, increased levels of ubiquitination were observed in NEK2-deficient conditions (Fig. 5j, k). Collectively, these results suggest that NEK2 controls the stability of PD-L1 in a kinase-dependent manner. As demonstrated by Hung's group, glycogen synthase kinase 3β (GSK3β) is an additional key protein that interacts with PD-L1 and induces phosphorylation-dependent proteasome degradation of PD-L1[15]. Thus, the relationship between the role of NEK2 and GSK3β in regulating the expression level of PD-L1 was further explored. Interestingly, our results revealed that NEK2 inhibitor reduced the phosphorylation of GSK3β, but reversed the up-regulation of PD-L1 caused by the down-regulation of p-GSK3β, and affected the PD-L1–GSK3β interaction (Supplementary Fig. 9a–d). In contrast, the effect of the GSK3β inhibitor on NEK2 and PD-L1 expression was not very stable and significant in pancreatic cancer cells (Supplementary Fig. 9e, f). These results suggest that NEK2 plays a more important role than GSK3β in the regulation of PD-L1, at least in pancreatic cancer.

**NEK2 maintains the stability of PD-L1 by phosphorylation at T194/T210.** In addition to upregulation of PD-L1 expression, its glycosylation at the N192/N200/N219 residues was also stimulated by NEK2 overexpression (Supplementary Fig. 10a–c). The influence of glycosylation on PD-L1 stability can be regulated by kinase-mediated phosphorylation, suggesting NEK2 controls the protein level of PD-L1 in such a manner. Two evolutionarily conserved NEK phosphorylation-specific motifs (F/LXXS/T) were identified across multiple species, centering at the T194 and T210 residues (Mus musculus: T193 and T209) in the glycosylation-rich region (N192/N200/N219) (Fig. 6a). Specific antibodies designed to recognize PD-L1 phosphorylated at T194 and T210 (T193/T209 in the mouse) were specifically generated to detect potential modification of NEK2 on PD-L1 (Fig. 6b). A significant decrease in T193/T209 phosphorylation levels of PD-L1 was observed in the presence of the NEK2 inhibitor and in NEK2 KD cells (Fig. 6c). In vitro kinase assay demonstrated that NEK2 directly phosphorylated PD-L1 (Fig. 6d). To further investigate the influence of NEK2-mediated phosphorylation on PD-L1 expression levels, constitutively non-phosphorylated (T193/209A) and phosphorylated (T193/209D) mutants of PD-L1, in addition to a kinase-dead (K37R) mutant of NEK2, were individually constructed. The results demonstrated that the T193/209A mutation in PD-L1 reduced its expression but the T193/209D mutation increased its expression in pancreatic cancer cells, while

the K37R mutation in NEK2 caused its loss of regulatory effect on the expression and phosphorylation of PD-L1 (Fig. 6e–g). Moreover, a shortened or extended half-life of PD-L1 was observed with T193/209A and T193/209D mutants, respectively (Supplementary Fig. 11a–d), while no effect is observed with the NEK2 K37R mutant on PD-L1 half-life (Supplementary Fig. 11e, f). Furthermore, increased and decreased ubiquitination of PD-L1 were observed in PD-L1 T193/209A and T193/209D mutants, respectively (Fig. 6h), while no influence was consistently observed with the NEK2 K37R mutant on PD-L1 ubiquitination (Fig. 6i). Taken together, the results demonstrate that NEK2 maintains PD-L1 stability through phosphorylation at the T193/T209 residues.

**NEK2 inhibition sensitizes PD-L1-targeted pancreatic cancer immunotherapy.** Since NEK2 regulates the stability of PD-L1, we hypothesized that inhibition of NEK2 may enhance the therapeutic efficacy of PD-L1-targeted drugs. Indeed, NEK2 inhibitor plus anti-PD-L1 antibody significantly improved the T cell-mediated cytotoxic effect in vitro compared with mono-NEK2 inhibitor or mono-anti-PD-L1 antibody (Supplementary Fig. 12a, b). In light of this, we additionally administrated NEK2 inhibitor and/or anti-PD-L1 antibody to KPC-bearing mice (Fig. 7a), after which both tumor volumes (Fig. 7b, c) and tumor weight (Fig. 7d) decreased significantly in mice that received the combined treatment. Additionally, the combination of NEK2 inhibitor and anti-PD-L1 antibody did not influence the weight of the mice (Fig. 7e). Additional flow cytometric analysis revealed a significant elevation in the quantity and activation of T cells infiltrated into the tumors in which both therapies were administered (Fig. 7f, g). Moreover, the combinatorial strategy was also evaluated in an orthotopic model (Supplementary Fig. 13a), where a similar outcome was observed (Supplementary Fig. 13b–d). Additionally, it was confirmed that the NEK2 inhibitor functionally inhibited NEK2 activity in mice (Supplementary Fig. 13e). Collectively, the results suggest that the NEK2 inhibitor enhanced the efficacy of PD-L1 blockade both in vitro and in vivo, alleviating immuno-resistance in pancreatic cancer.

**Discussion**
Increasing evidence suggests that conventional mediators of the cell cycle are also involved in cancer immune regulation. For instance, cyclin-dependent kinase 4, 5, and 6 (CDK4/5/6), three serine/threonine kinases that are conserved throughout eukaryotes, act as the fundamental driving forces for cell cycle progression and thus play an important role in the occurrence and development of multiple malignant tumors[31,32]. Intriguingly, it has been reported that CDK4/6 inhibitors not only suppress tumor cell growth by inducing cell cycle arrest, they also trigger tumor immune evasion through inhibition of cullin 3–SPOP E3 ligase-mediated proteasomal degradation of PD-L1 that increases

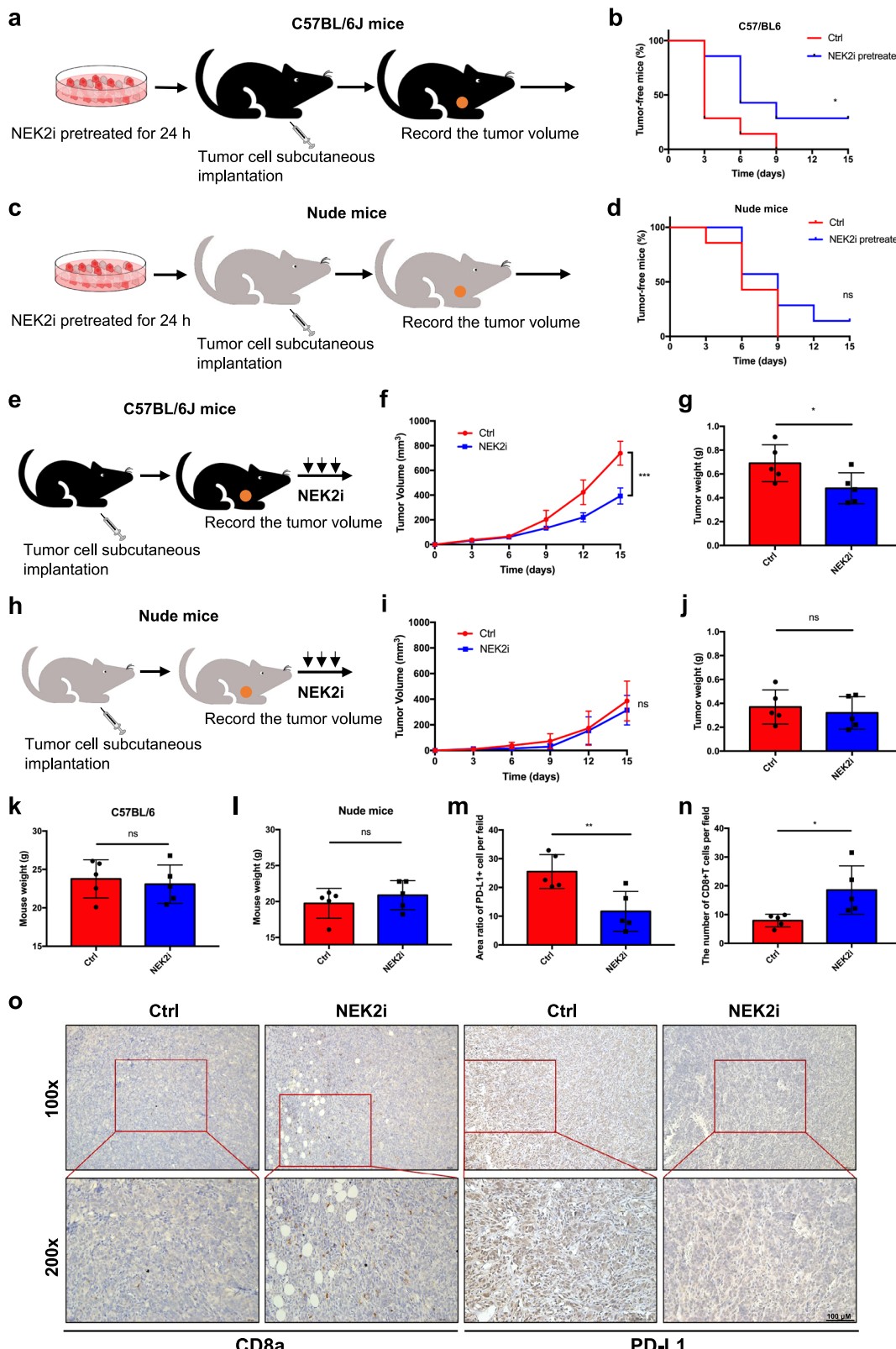

the abundance of the protein[33]. Furthermore, combination therapy of CDK4/6 inhibitors and PD-1/PD-L1 blockade antibodies improve the efficacy of immunotherapy in multiple cancer types[32–35]. CDK5 regulates the expression of PD-L1 via the interferon regulatory factors IRF2 and IRF2BP2 in medulloblastoma[36]. As mentioned above, NEK2 is a multifunctional protein involved in non-immune and immune function including cell cycle regulation, microtubule stabilization, kinetochore attachment, spindle assembly checkpoint, phosphorylation of downstream proteins, and the maintenance of normal development and function of B cells, and has been associated with tumor progression and clinical prognosis in multiple cancers. In the present study, NEK2 has been identified as another critical regulatory factor in cancer immune resistance.

**Fig. 3 NEK2 inhibition enhances the anti-pancreatic cancer immune response. a, c** Schematic protocols of pancreatic cancer cells with or without pretreatment with a kinase-specific inhibitor of NEK2 (10 μM, 24 h) separately and s.c. injected into immunocompetent and immunodeficient mice ($n = 7$). **b, d** Tumor incidence in immunocompetent and immunodeficient mice was recorded at the times indicated. Tumors were further treated using an NEK2 inhibitor (100 μg/mouse, 2 weeks). **e, f** Schematic protocol of the treatment schedule and tumor growth curve in immunocompetent mice ($n = 5$). Tumors were measured at the time points indicated and removed at the endpoint. (**h, i**) Schematic protocol of the treatment schedule and tumor growth curve in immunodeficient mice ($n = 5$). Tumors were measured at specified time points and dissected at the endpoint. Tumor and mouse weight of immunocompetent (**g, k**) ($n = 5$) and immunodeficient mice (**j, l**) ($n = 5$) as reported at the endpoint. **m–o** Representative images and further quantification of tumor-infiltrating lymphocytes. Scale bars: 100×: 50 μm; 200×: 100 μm. Kaplan–Meier method and a Gehan–Breslow–Wilcoxon test are indicated in **b** and **d**. Results represent means ± SD of one representative experiment in **f, g, i, j, k, l, m**, and **n**. All data are representative of three independently performed experiments. *$P < 0.05$, **$P < 0.01$, ***$P < 0.001$ using a two-tailed $t$-test; ns: not significant.

Overall, these findings provide a theoretical basis for the development of an anti-cancer treatment strategy combining cell cycle-related kinase inhibitors with PD-1/PD-L1-targeted immunotherapy.

PD-L1 is expressed in a variety of cells, especially cancer cells and macrophages, and is critical for tumors to escape immune surveillance[37]. To date, PD-1/PD-L1 blockade has become the most successful strategy in cancer immunotherapy against solid tumors[38]. However, the overall response rate for PD-1 or PD-L1 blockade rarely exceeds 40%[39]. One reason that accounts for this phenomenon is the PTM of PD-L1, which potentially impacts the therapeutic efficacy of PD-L1 blockade. The mechanism of PD-L1 PTM has been extensively investigated and multiple regulators affecting PD-L1 stability via PTMs have been identified, providing methods to improve the efficacy of PD-L1 blockade[15,40–42]. For instance, EGF signaling induces PD-L1 glycosylation, while Gefitinib (an EGFR inhibitor) sensitizes PD-L1 blockade therapy in a syngeneic mouse model[15]. Based on these findings, the present study further demonstrated that NEK2 is a critical regulator of PD-L1 phosphorylation, and is involved in tumorigenesis and the progression of pancreatic cancer. NEK2 directly interacts with and phosphorylates PD-L1 to maintain its stability. The combination of a NEK2 inhibitor and anti-PD-L1 antibody synergistically suppresses tumor growth both in vivo and in vitro, thus representing the potential for sensitizing PD-L1-targeted therapy in pancreatic cancer.

Given the considerable influence of the tumor immune microenvironment on immune checkpoint blockade (ICB) therapy, we investigated the prognostic roles of NEK2 in pancreatic cancers with different immune statuses. NEK2 was significantly correlated with the prognosis of patients of immunologically "hot" status, while no significant correlation was observed in immunologically "cold" PDAC. The presence of tumor-killing T cells demonstrates the capability of the immune system to destroy tumors and its sensitivity to ICB[43]. PDAC is characterized by substantial immunological heterogeneity and differential levels of T cell infiltration[44–46]. "Hot" tumors typically respond well to checkpoint treatments[47–49]. Hence, NEK2 may be considered a therapeutic biomarker for ICB, and intervention with NEK2 may enhance the efficacy of PD-L1 blockade in "hot" pancreatic tumors. In addition to T cells, a great number of immunosuppressive populations are present in PDAC, also greatly influencing the efficacy of checkpoint therapy. Interestingly, the expression of both NEK2 and PD-L1 was also observed in macrophages, DCs, and MDSCs in addition to tumor cells[37]. In light of this, we speculated that NEK2 may also regulate the expression of PD-L1 in these suppressive populations. TMB is a measure of the number of somatic mutations in tumors and can be treated as a biomarker that reflects the potential therapeutic efficacy of immunotherapy, allowing the identification of patients who may benefit[50]. A high mutation load correlates with increased numbers of neoantigens, leading to T cell immunoreactivity and sensitivity to ICB treatments[51,52]. Interestingly, the

expression levels of NEK2 can be used to predict the prognosis of patients with low TMB, suggesting that inhibition of NEK may be a promising strategy for increasing the effectiveness of immunotherapy in tumors with low TMB.

The effect of phosphorylation on the regulation of PD-L1 stability is complex, indirect, and sometimes even contradictory, but ultimately closely associated with modification by glycosylation. Recently, Hung's group revealed that IL-6-activated JAK1 phosphorylates PD-L1 at Tyr112, catalyzing PD-L1 glycosylation through recruitment of endoplasmic reticulum-associated N-glycosyltransferase STT3A and so maintaining its stability[53]. Glycosylation of PD-L1 at N192, N200, and N219 contributes to PD-L1 stability and is important for its immunosuppressive function. Therefore, it is possible to generate a therapeutic antibody (STM108-ADC) that specifically recognizes the N192 and N200 glycosylation sites of PD-L1[15,42]. However, it was revealed in another study that phosphorylation of PD-L1 S195 can induce abnormal glycosylation and prevent the translocation of PD-L1 to the cell membrane and subsequently, PD-1 binding, increasing the activity of CTL[16]. In the present study, we identified the T194/T210 residues of PD-L1 as targeting sites for NEK2-mediated phosphorylation. Since the newly revealed T194 site is close to S195, its phosphorylation may interrupt S195 modification, potentially promoting normal PD-L1 glycosylation, thus increasing its stability. Nonetheless, specific drugs that target the T194/T210 sites of PD-L1 and analysis of their specificity and clinical safety remain to be defined in the future.

In conclusion, NEK2 is a prognostic factor of immunologically "hot" and low mutational pancreatic cancer. NEK2 interacts with and phosphorylates PD-L1 at the T194/T210 residues, maintaining its stability, and thus inhibition of NEK2 leads to expedited degradation of PD-L1, activating cytotoxic T cells to obliterate pancreatic cancer cells (Fig. 8). Consequently, while further preclinical studies and clinical trials are warranted for demonstration, NEK2 is nevertheless a promising target for pancreatic cancer therapy, and the combinatorial inhibition of NEK2 and PD-L1 will synergistically benefit pancreatic cancer patients.

## Methods

**Antibodies.** Antibodies were obtained from the following sources: anti-NEK2 (sc-55601, Santa Cruz, 1:200; 24171-1-AP, Proteintech Group, 1:2000; 610594, BD biosciences, 1:100), anti-Phospho-NEK2 (Ser170) (PA5-105332, Invitrogen, 1:1000), anti-PD-L1 (13684, Cell Signaling Technology, 1:100; ab213480, Abcam, 1:1000; ab205921, Abcam, 1:1000; 66248-1-Ig, Proteintech Group, 1:2000; 14-5983-82, Thermo Scientific, 1:100), anti-Ubiquitin (3933, Cell Signaling Technology,1:1000), anti-CD8a (98941, Cell Signaling Technology, 1:400), anti-GSK-3β (12456, Cell Signaling Technology, 1:1000), anti-phospho-GSK-3β (Ser9) (5558, Cell Signaling Technology, 1:1000), anti-IRE1α (3294, Cell Signaling Technology, 1:1000), anti-HSP90B1 (NBP2-42379, Novus Biologicals, 1:500), anti-EGFR (4267T, Cell Signaling Technology, 1:1000), anti-GST (2624, Cell Signaling Technology, 1:1000), anti-β-actin (Ab8226, Abcam, 1:2000), anti-GAPDH (AF5009, Beyotime, 1:1000), anti-α-tubulin (AF0001, Beyotime, 1:2000), anti-Flag (2368, Cell Signaling Technology, 1:1000), anti-GFP (2956, Cell Signaling Technology, 1:1000), rabbit IgG isotype control (8726, Cell Signaling Technology, 1:100), mouse IgG isotype control (sc-2025, Santa Cruz, 1:200), HRP goat anti-

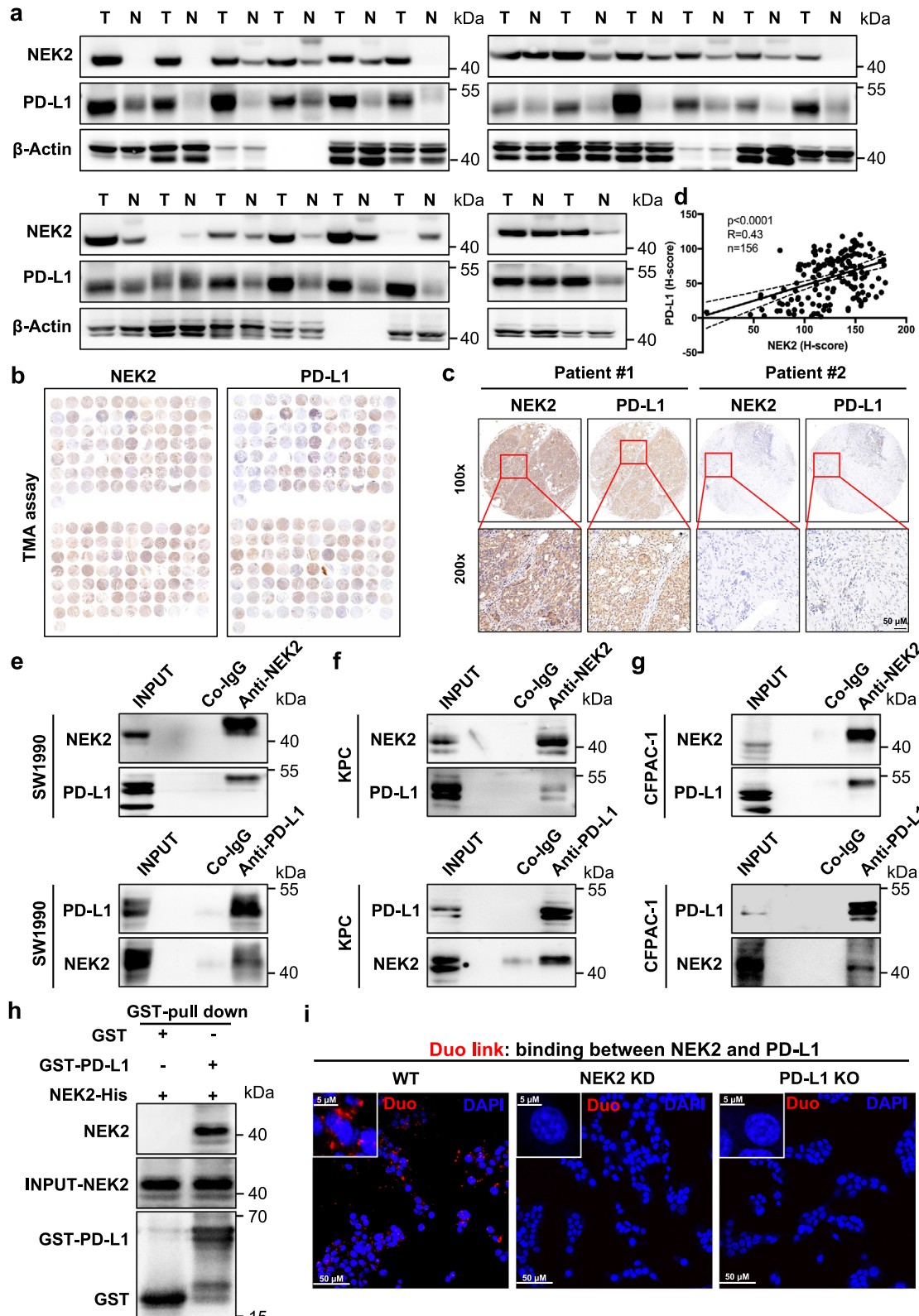

**Fig. 4 NEK2 positively correlates and interacts with PD-L1 in pancreatic cancer. a** Western blot analysis of NEK2 and PD-L1 in clinical pancreatic tissue samples from patients ($n = 20$) (N: Normal pancreatic tissue; T: Pancreatic tumor tissue). **b–d** Representative images and statistical results of IHC staining of NEK2 and PD-L1 in a tissue microarray ($n = 156$). **e–g** Cell lysates from SW1990, KPC, and CFPAC-1 separately analyzed by IP and Western blotting using the antibodies indicated. Representative image is shown $n = 3$ independent experiments. **h** GST-pull down assay of NEK2-His and GST-PD-L1 protein. Representative image is shown $n = 3$ independent experiments. **i** Representative images of individual immunofluorescence staining of NEK2 and PD-L1 interaction in KPC cells by Duolink assay. The red dots (NEK2/PD-L1 interaction) indicate their interaction. Representative image is shown $n = 3$ independent experiments. The Spearman correlations and *p*-values by Spearman's test are indicated in **d**.

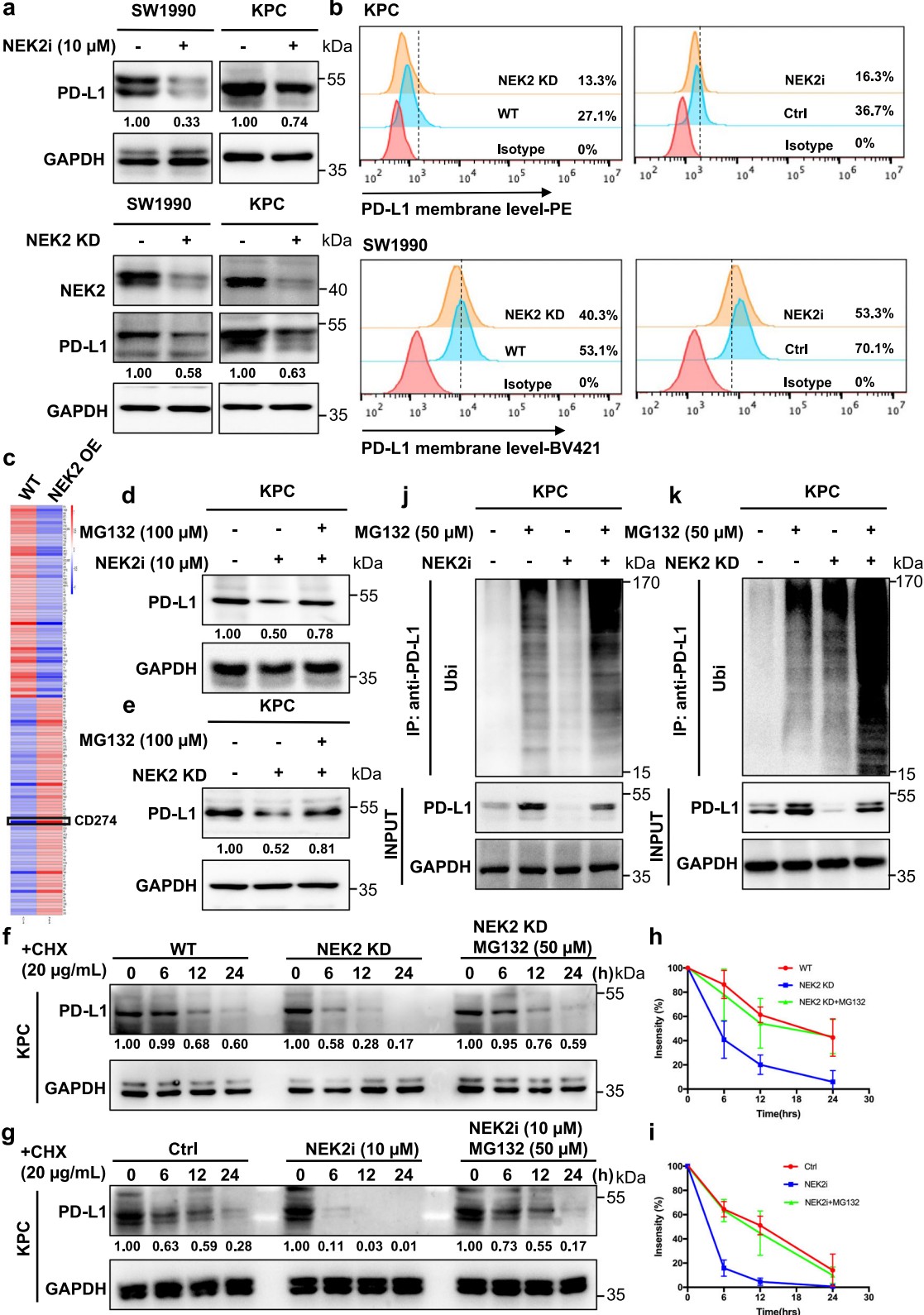

rabbit IgG (A0208, Beyotime, 1:5000 for WB, 1:100 for IHC), HRP goat anti-mouse IgG (A0216, Beyotime, 1:5000 for WB, 1:100 for IHC), goat anti-rabbit IgG (GTX77061, GeneTex, 1:2000), goat anti-mouse IgG (GTX26708, GeneTex, 1:2000), FITC anti-mouse CD3 (100203, Biolegend, 1:200), APC/Cy7 anti-mouse CD3 (100329, Biolegend, 1:200), Brilliant Violet 785 anti-mouse CD45 (304048, Biolegend, 1:200), PE/Cy7 anti-mouse CD8a (100722, Biolegend, 1:200), Brilliant Violet 605 anti-mouse CD8a (100744, Biolegend, 1:200), FITC anti-mouse IFN-γ (505806, Biolegend, 1:200), Brilliant Violet 421 anti-human/mouse Granzyme B

Recombinant (396414, Biolegend, 1:200), PE anti-mouse Perforin (154306, Biolegend, 1:200), APC anti-mouse TNF-α (506308, Biolegend, 1:200), BV785 anti-human CD45 (304047, Biolegend, 1:200), PE/Cy7 anti-human CD11b (301322, Biolegend, 1:200), FITC anti-human CD68 (333806, Biolegend, 1:200), PerCP/Cyanine5.5 anti-human HLA-DR (307630, Biolegend, 1:200), APC/Cy7 anti-human CD11c (337218, Biolegend, 1:200), APC anti-human CD15 (301908, Biolegend, 1:200), PE anti-mouse CD274 (124308, Biolegend, 1:200), Brilliant Violet 421 anti-human CD274 (329714, Biolegend, 1:200), PE rat IgG2b, κ isotype control

**Fig. 5 NEK2 inhibits ubiquitination-mediated proteasomal degradation of PD-L1. a, b** Western blot analysis and flow cytometry of PD-L1 expression in pancreatic cancer cell lines after treatment with NEK2 inhibitor (10 μM, 24 h) and *NEK2* knockdown. Representative image is shown $n = 3$ independent experiments. **c** LC–MS proteomics quantitative analysis of pancreatic cancer cells overexpressing *NEK2*. **d** Western blot analysis of PD-L1 expression in KPC cells treated with NEK2 inhibitor (10 μM, 24 h) after treatment with MG132 (100 μM, 24 h). Representative image is shown $n = 3$ independent experiments. **e** Western blot analysis of PD-L1 expression in WT and NEK2 KD KPC cells treated with MG132 (100 μM, 24 h). Representative image is shown $n = 3$ independent experiments. **f** Stability analysis of PD-L1 in KPC cells treated with NEK2 inhibitor (10 μM, 24 h) after treatment with cycloheximide (CHX) (20 μg/mL). Representative image is shown $n = 3$ independent experiments. **g** Stability analysis of PD-L1 in WT and NEK2 KD KPC cells treated with CHX (20 μg/mL). Representative image is shown $n = 3$ independent experiments. **h, i** Statistical analysis of three independent experiments is displayed. The intensity of PD-L1 protein expression was quantified using a densitometer. **j** Ubiquitination assay of PD-L1 in KPC cells treated with NEK2 inhibitor (10 μM, 24 h), subjected to anti-PD-L1 IP and anti-ubiquitin Western blot analysis after treatment with MG132 (50 μM, 24 h). **k** Ubiquitination assay of PD-L1 in WT and NEK2 KD KPC cells treated with MG132. Results represent means ± SD of one representative experiment in **h** and **i**. All data are representative of three independently performed experiments. \*$P < 0.05$, \*\*$P < 0.01$, \*\*\*$P < 0.001$ using a two-tailed t-test; ns: not significant.

---

(400607, Biolegend, 1:200), Brilliant Violet 421 mouse IgG2b, κ isotype control (400342, Biolegend, 1:200), PE rat anti-mouse IgG1 (550083, BD biosciences, 1:200), Trustain FcX anti-mouse CD16/CD32 (101320, Biolegend, 1:200), human TruStain FcX (422302, Biolegend, 1:200), and InVivoMAb anti-mouse PD-L1 (BE0101, BioXcell).

**Agents, kits, and plasmids**. Chemicals, kits and agents were acquired from designated suppliers: NCL00017509 (5150, Tocris), TWS119 (S1590, Selleck), Cycloheximide (239763-M, Sigma-Aldrich), MG132 (M8699, Sigma-Aldrich), Puromycin (ant-pr-1, Invivogen), cell lysis buffer for Western or IP (P0013, Beyotime), protease inhibitor cocktail (B14001, Bimake), phosphatase inhibitor cocktail (B15001, Bimake), protein A/G Dynabeads (B23201, Bimake), leukocyte activation cocktail (550583, BD Biosciences), Percoll (17-0891-01, GE Healthcare), collagenase IV (17104019, Thermo Fisher Scientific), dispase (17105041, Gibco), DNase (D5025, Sigma-Aldrich), calcium chloride solution (21115, Sigma-Aldrich), Lipofectamine 3000 (L3000-075, Life Technologies), Ultracruz transfection reagent (sc-395739, Santa Cruz), Plasmid Transfection Medium (sc-108062, Santa Cruz), Fixation/Permeabilization solution kit (555028, BD Biosciences), DAB Chromogen kit (BDB2004, Biocare), Duo-link kit (DUO92101, Sigma-Aldrich), ER enrichment kit (NBP2-29482, Novus Biologicals), Pierce GST Protein Interaction Pull-Down Kit (21516, Thermo Scientific), LIVE/DEAD™ Fixable Violet dead cell stain kit with 405 nm excitation (L34955, Thermo Fisher Scientific), MidiMACS™ separator and starting kit (130-042-301, Miltenyibiotec), and CD8a+ T cell isolation kit, mouse (130-096-495, Miltenyibiotec); human NEK2 double nickase plasmids (sc-417360-NIC, Santa Cruz), mouse Nek2 double nickase plasmids (sc-421853-NIC, Santa Cruz), NEK2 cDNA ORF clone, human, C-Myc tag plasmid (HG10054-CM, Sinobiological), human CD274 (T194A/T210A) plasmid (Shanghai OBiO Technology), human CD274 (T194D/T210D) plasmid (Shanghai OBiO Technology), mouse Cd274 (T193A/T209A) plasmid (Shanghai OBiO Technology), mouse Cd274 (T193D/T209D) plasmid (Shanghai OBiO Technology), and mouse NEK2 (K37R) plasmid (Shanghai OBiO Technology).

**Cell culture**. The KPC cell line, derived from spontaneous tumors in a Kras$^{LSL-G12D}$; Trp53$^{LSL-R172H}$; Pdx1-Cre mouse model, was a kind gift from the laboratory of Prof. Raghu Kalluri (MD Anderson Cancer Center, Houston, TX, USA). All other PDAC cell lines were purchased from ATCC (American Type Culture Collection). The KPCs were maintained in McCoy's 5A (Modified) Medium (Thermo Fisher Scientific). All other PDAC cell lines used in the present study were maintained in 1640 medium (GE Healthcare Life Sciences SH30027.0) supplemented with 10% FBS (Biological Industries 04-001-1) and 1% Penicillin/Streptomycin (Cienry CR-15140). The presence of mycoplasma contamination was routinely evaluated in all cultures by PCR. All the cell lines are routinely checked for morphological and growth changes to probe for cross-contaminated, or genetically drifted cells. If any of these features occur, we use the short tandem repeat (STR) profiling service by ATCC to re-authenticate the cell lines.

**Animal care**. Six-week-old C57BL/6 and Balb/c nude mice were purchased from the Model Animal Research Center of Nanjing University and maintained in a specific-pathogen-free (SPF) environment in the Experimental Animal Center, the First Affiliated Hospital, School of Medicine, Zhejiang University. All animal experiments were approved by the Ethics Committee of the First Affiliated Hospital, School of Medicine, Zhejiang University. Animal suffering was minimized or prevented at all times to improve their welfare.

**Animal procedures**. KPC cells were seeded at a density of $2 \times 10^5$ cells in 50 μl serum-free DMEM with or without pretreatment with the NEK2 inhibitor (10 μM) (Tocris Bioscience, NCL 00017509, potent and reversible NIMA related kinase 2 (Nek2) inhibitor, 3-[(6-Ethynyl-9H-purin-2-yl) amino]benzeneacetamide[54]) for 24 h. The cells were separately injected s.c. into the right flank of 6–8-week-old male C57BL/6 mice and nude mice ($n = 7$). The incidence of tumors was recorded. KPC-NEK2 KD cells and KPC-WT cells were both seeded at a density of $5 \times 10^5$ in 50 μl serum-free

DMEM, then separately injected by s.c. into the right flanks of 6–8-week-old male C57BL/6 mice and nude mice ($n = 7$). Tumor growth was measured using a caliper, and their size was recorded. After the experiments, tumors in the C57BL/6 mice were harvested and divided into three groups. One-third were lysed to obtain a single cell suspension of tumor cells that were analyzed by FACS. One-third was snap-frozen in liquid nitrogen for further PCR and immunoblot analysis. The remaining third was fixed in 10% neutral buffered formalin, paraffin-embedded then sectioned for additional IHC pathological analysis. Tumors in nude mice were harvested and used for PCR, immunoblotting, and pathological analysis. Combination therapy in the in vivo experiments was accomplished using a subcutaneous model, in which KPC cells seeded at a density of $5 \times 10^5$ in 50 μl serum-free DMEM were administered by s.c. injection into the right flank of 6–8-week-old male C57BL/6 mice. Treatments started when tumors reached a volume of 50–100 mm³. Tumor-bearing mice were randomly divided into four groups and treated with NEK2 inhibitor (100 μg/mouse) or anti-PD-L1 (200 μg/mouse), either individually or in combination ($n = 5$), three times every week for 2 weeks, plus an untreated control. InVivoMAb anti-mouse PD-L1 (Bio X Cell, West Lebanon, NH, USA) and IgG isotype control (Bio X Cell, West Lebanon, NH, USA) were administered to the mice via intraperitoneal injection and the NEK2 inhibitor administered via intratumoral injection. Prior to each administration, tumor growth was measured using calipers, and the size was recorded. Following the experiments, the tumors were harvested and divided into three groups, as above. Combination therapy in the in vivo experiments was also accomplished using an orthotopic model, in which a small incision was created in the left abdomen close to the spleen. The pancreas was located in front of the right side of the spleen. KPC cells at a density of $5 \times 10^5$ in 20 μl medium mixed with 10 μl Matrigel were injected into the pancreas using a sterile insulin needle. Treatment began when tumors reached a size of 50–100 mm³, as measured by in vivo imaging. Tumor-bearing mice were randomly divided into four groups then treated with an intraperitoneal injection of NEK2 inhibitor (200 μg/mouse) and anti-PD-L1 (200 μg/mouse), either individually or in combination ($n = 5$), three times every week for 2 weeks, plus an untreated control. Following the experiment, the tumors were harvested and divided into three groups, as described above. For survival experiments, KPC-NEK2 KD cells and KPC-WT cells, both at a density of $5 \times 10^5$ in 20 μl medium mixed with 10 μl Matrigel were injected into the pancreas using a sterile insulin needle. The time point of death of each mouse was tracked and recorded, and a survival curve was plotted. Tumor volume was calculated from measurements of the longest diameter and the shortest diameter perpendicular to it, as follows: $1/2 \times length \times width^2$.

**Human tissues**. Human pancreatic adenocarcinoma cancer tissue specimens were obtained from the Department of Hepatobiliary and Pancreatic Surgery, the First Affiliated Hospital, School of Medicine, Zhejiang University. The protocol was approved by the Institutional Review Board at the First Affiliated Hospital, School of Medicine, Zhejiang University. Written informed consent was obtained from each patient at the time of enrollment. Tissue microarray slides with 156 paraffin-embedded patient PDAC tissue samples were prepared with the assistance of Wuhan Servicebio Technology.

**Cell transfection and generation of stable NEK2 knockdown cells**. Cells at 70% confluence were transiently transfected with human *NEK2*/mouse *Nek2* DNA via a human *CD274* (T194A/T210A)/(T194D/T210D) plasmid, or mouse *Cd274* (T193A/T209A)/(T193D/T209D) plasmid using Lipofectamine 3000 (Life Technologies, Carlsbad, CA, USA). The efficiency of transfection was determined using Western blotting and RT-PCR to evaluate protein and mRNA expression following cell collection. KPC and SW1990 cells were stably transfected with human *NEK2*/mouse *Nek2* double nickase plasmids (Santa Cruz Biotechnology, Dallas, TX, USA) using Ultracruz transfection reagent (Santa Cruz Biotechnology, Dallas, TX, USA), in accordance with the manufacturer's instructions. Cells were cultured until 50% confluent, then 12 h before transfection the medium was replaced with plasmid transfection medium (Santa Cruz Biotechnology, Dallas, TX, USA) containing the appropriate plasmids. After incubation for 24 h, the medium was replaced with a fresh medium, and the transfected cells selected after treatment with 10 μg/ml puromycin (InvivoGen, San Diego, CA, USA) for one week.

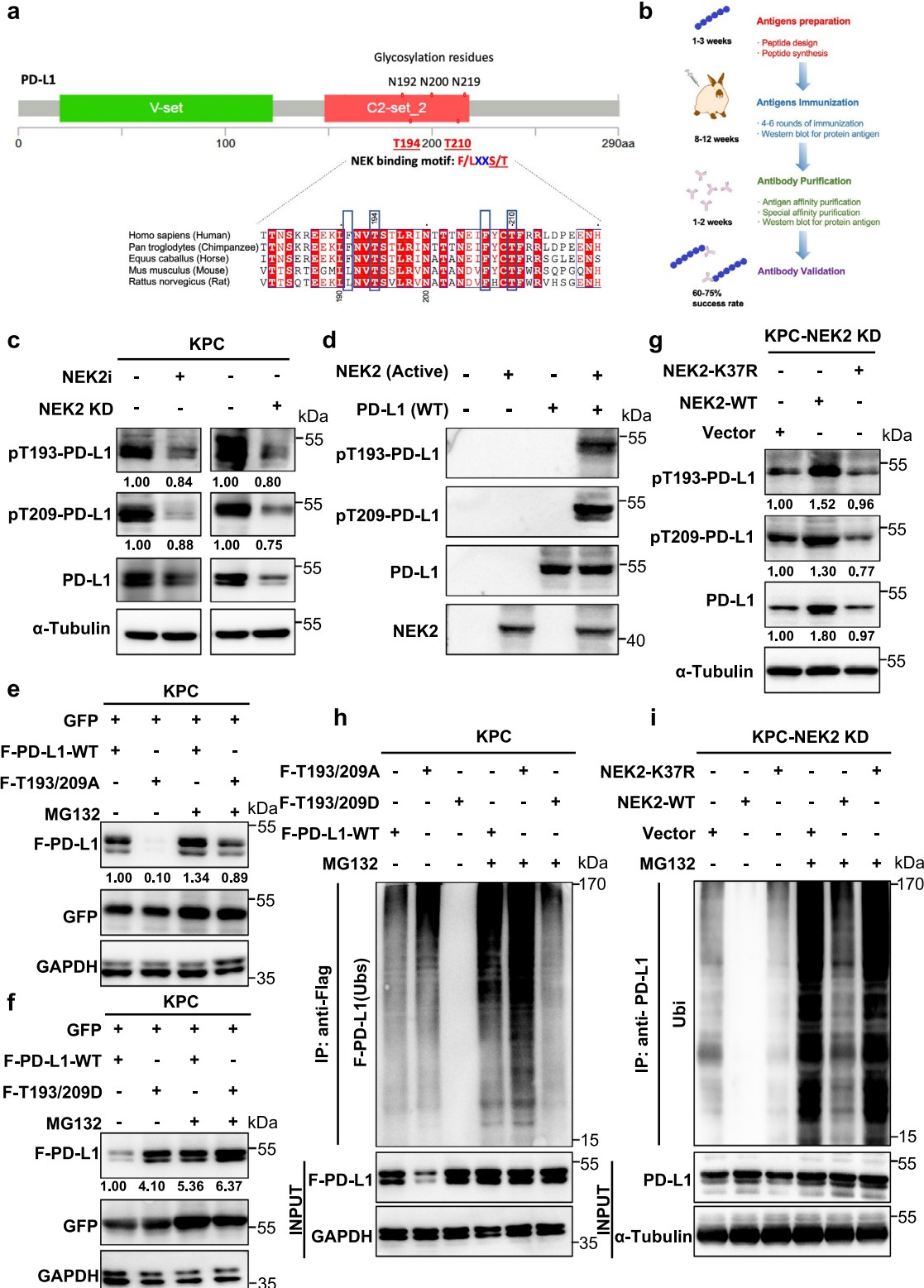

**T cell-mediated tumor cell destruction assay**. CD8+ T cells were isolated from the mouse spleen using a CD8+ T cell isolation kit, an LS Column, and a MidiMACS™ Separator. Isolated CD8+ T cells were fluorescently stained using CD8-PE and CD3-FITC then analyzed using flow cytometry to confirm their phenotype. KPCs both untreated and treated with NEK2 inhibitor (10 µM) for 24 h were incubated for 48 h with activated CD8+ T cells. KPC-NEK2 KD cells and KPC-WT cells were incubated overnight and allowed to adhere to the culture plates, then treated with isolated activated CD8+ T cells and incubated for 48 h. KPC cells were incubated overnight to allow adhesion to the culture plates and then treated with isolated activated CD8+

T cells with NEK2 inhibitor (10 µM) and anti-PD-L1 (200 µg), either individually or in combination, then incubated for 48 h. The ratio of tumor cells to activated CD8+ T cells was 1:4. T cells and cell debris were removed by washing three times with PBS. The remaining living cancer cells were fixed with 4% paraformaldehyde, stained with 0.5% crystal violet, and quantified using a spectrophotometer at OD 570 nm.

**Immunoblotting and immunoprecipitation**. For immunoblotting, cells were lysed using RIPA lysis buffer (P0013B Beyotime Biotechnology) containing

**Fig. 6 NEK2 phosphorylates PD-L1 at T194/T210 to maintain protein stability. a** Schematic diagram of the NEK-binding motif (F/LXXS/T) in the glycosylation-rich region and amino acid sequences around the potential binding sites of PD-L1 were aligned in evolutionarily divergent species. The F/LXXS/T motifs are highlighted in blue. **b** Generation of site-specific antibodies against T194 and T210 (Mus musculus: T193 and T209)-phosphorylated PD-L1. **c** Western blot analysis of T193 and T209-phosphorylated PD-L1 in KPC cells with NEK2 inhibitor (10 μM, 24 h) and NEK2 KD KPC. Representative image is shown $n = 3$ independent experiments. **d** In vitro kinase assay and western blot analysis of pT193-PD-L1 and pT209-PD-L1 expression of recombinant PD-L1 WT and NEK2 (active) protein. Representative image is shown, $n = 3$ independent experiments. **e**, **f** Western blot analysis of PD-L1 expression in flag-PD-L1 WT and T193/209A or T193/209D-transfected KPC cells with or without MG132 treatment. Representative image is shown $n = 3$ independent experiments. **g** Western blot analysis of pT193-PD-L1, pT209-PD-L1, and PD-L1 expression in WT and K37R transfected NEK2 KD KPC cells. Representative image is shown $n = 3$ independent experiments. **h** Ubiquitination assay of PD-L1 in Flag-PD-L1 WT and T193/209A or T193/209D-transfected KPC cells, subjected to anti-PD-L1 IP and anti-ubiquitin Western blot analysis after treatment with MG132 (50 μM, 24 h). Representative image is shown $n = 3$ independent experiments. **i** Ubiquitination assay of PD-L1 in WT and K37R transfected NEK2 KD-KPC cells subjected to PD-L1 IP and Western blot analysis with anti-ubiquitin after treatment with MG132 (50 μM, 24 h). Representative image is shown $n = 3$ independent experiments.

phenylmethanesulfonyl fluoride (ST505 Beyotime Biotechnology) for 30 min on ice, after which the mixture was centrifuged at 12,000×*g* for 15 min and the upper layer containing soluble proteins collected. Protein concentration was measured using a bicinchoninic acid (BCA) reagent (P0012 Beyotime Biotechnology). Lysates were heated to 100 °C in NuPAGE LDS sample buffer (4×) (Thermo Fisher Scientific) for 3–5 min. Proteins were separated by SDS–PAGE then transferred to a PVDF membrane (Millipore) which was blocked in 5% milk in TBST, probed with a panel of antibodies, and corresponding bands visualized using a ChemiScope-Touch (Clinx Science Instruments). For co-immunoprecipitation (IP) experiments, cells were lysed in IP/Western lysing solution (P0013 Beyotime Biotechnology) containing a phosphatase inhibitor cocktail (B15001 Bimake) and a protease inhibitor cocktail (B14001 Bimake) for 60 min on ice, then centrifuged at 12,000×*g* for 15 min to remove debris. Cleared lysates were analyzed by IP with the appropriate antibodies. Lysates were incubated with a primary antibody for 4–6 h, then protein A or protein G Dynabeads (B23201 Bimake) were added and incubated for 2–4 h at 4 °C. After washing three times in washing buffer (10% IP/Western lysing buffer), the samples were heated to 100 °C in NuPAGE LDS Sample Buffer (4×) (Thermo Fisher Scientific) for 3–5 min, then immunoblotted as described above. The intensity of immunoblot bands was evaluated using ImageJ 1.8.0 software (National Institutes of Health, Bethesda, USA).

**Quantitative real-time reverse transcription–PCR (qRT-PCR) analysis.** Total RNA was extracted from tissues or cells using Trizol LS Reagent (Invitrogen, Carlsbad, CA, USA). Samples were washed three times with PBS, then RNA concentration measured using a Thermo Scientific™ NanoDrop™ One instrument. RNA was reverse transcribed into cDNA using a PrimeScript RT reagent kit (RR047A, Takara) in accordance with the manufacturer's instructions. qRT-PCR was performed using a 20 μL reaction system in a real-time PCR machine (Applied Biosystems 7500 Fast Real-Time PCR System, Applied Biosystems) with TB Green Premix Ex Taq™ II (RR820A, Takara). The relative expression of genes was normalized to *GAPDH*/*Gapdh* and calculated using the standard $2^{-\Delta\Delta Ct}$ method[55]. In addition to commercial primers, including those for human GAPDH (Shanghai Sangon Biotech, B661104), mouse Gapdh (Shanghai Sangon Biotech, B661304), human ACTB (Shanghai Sangon Biotech, B661102), and mouse Actb (Shanghai Sangon Biotech, B661302), additional primers were synthesized by Sunya Biotech (Hangzhou, China) as follows: human CD274 (Forward: TGGCATTTGCT-GAACGCATTT, Reverse: TGCAGCCAGGTCTAATTGTTTT); mouse Cd274 (Forward: GCTCCAAAGGACTTGTACGTG, Reverse: TGATCTGAAGGGCAG-CATTTC); human NEK2 (Forward: CGACGGTTAAACGGGGC, Reverse: TACAGCAAGCAGCCCAATGA); and mouse Nek2 (Forward: CTTGATCTTC-CATCCTCAGCCA, Reverse: CCTTGCGGTGTTCTCTTTGC).

**IHC.** Mouse tumors harvested at the end of the animal experiments were analyzed by IHC for positivity for CD8, NEK2, and PD-L1. Tissues were stored in 10% neutral buffered formalin, embedded in paraffin, sliced into 4 μm-thick sections, placed onto super frost+ glass slides, baked for 60–90 min at 68 °C, then deparaffinized. Antigens were retrieved by boiling the sections in sodium citrate antigen retrieval solution (Solarbio Life Science) for 10 min, after which they were incubated at room temperature for 25–30 min. Samples were blocked in 3% BSA for 30 min at room temperature, incubated overnight with a primary antibody at 4 °C, as appropriate, then with an HRP-conjugated secondary antibody for 50 min at room temperature. Target proteins were visualized using a diaminobenzidine (DAB) chromogen kit (BDB2004, Biocare), in which brown staining represented the targeted molecule. Slides were counterstained with diluted hematoxylin for 3–5 min. Representative images of each tumor were captured using ImageScope software (Leica Biosystems). Immunohistochemical staining of paraffin-embedded PDAC tissue microarray slides was performed by Wuhan Servicebio Technology, with antibodies against NEK2 (24171-1-AP, Proteintech Group) and PD-L1 (ab205921, Abcam). Staining performance was quantified by processing images using 3DHISTECH QuantCenter 2.1 software.

**Immunofluorescence staining.** To show the binding between NEK2 and PD-L1, pancreatic cancer cells were subjected to Duo-link assay (catalog DUO92101, Sigma-Aldrich) with anti-NEK2 (PA5-31259, Thermo Scientific) and anti-PD-L1 (14-5983-82, Thermo Scientific) as primary antibodies, according to the manufacturer's instruction. Fluorescent images of the cells were observed under a TCS SP8 X confocal microscope (Leica).

**Trypsinization of ER microsomal fractions.** ER fraction was collected from pancreatic cancer cells by ER enrichment kit (NBP2-29482, Novus Biologicals). After pretreatment with or without 1%, Triton X-100 for 3 min, the solution (0.625 g/L trypsin + 0.05 g/L EDTA in PBS) was added to ER fraction. Samples were incubated for the indicated time. After trypsinization, samples were analyzed by Western blotting with primary antibodies against IRE1α (3294, Cell Signaling Technology), HSP90B1 (NBP2-42379, Novus Biologicals), EGFR (4267T, Cell Signaling Technology), NEK2 (24171-1-AP, Proteintech Group), and a-Tubulin (AF5012, Beyotime).

**GST pull-down assay.** Commercial recombinant human GST (ab70456, Abcam), PD-L1/GST (ag12432; Proteintech), and NEK2-His (LS-G25896, LifeSpan Biosciences) were subjected to GST pull-down assay. The conjugation of GST and PD-L1/GST with glutathione beads and the pull-down assay were performed using Pierce GST Protein Interaction Pull-Down Kit (21516, Thermo Scientific) according to the manufacturer's instructions.

**In vitro kinase assay.** Recombinant PD-L1 WT (10084-H08H, sinobiological) were incubated with activated NEK2 (active) (14-545, Sigma-Aldrich) and 200 μM ATP (9804S, Cell Signaling Technology) in a kinase buffer (9802S, Cell Signaling Technology) at 30 °C for 30 min. The kinase reaction was stopped by the addition of SDS sample and boiling and subsequently subjected to Western blotting.

**Flow cytometric analysis and CTL profile analysis of mouse tumors.** Samples were mechanically dissociated into small fragments using scissors and scalpels, placed in DMEM supplemented with 2% FBS, collagenase IV (1 mg/ml) (17104019, Thermo Fisher Scientific), DNase (10 μg/ml) (D5025, Sigma-Aldrich), Dispase (0.6 mg/ml) (17105041, Gibco), and CaCl₂ (3 mM) (21115, Sigma-Aldrich), then incubated at 37 °C while shaking at 200 rpm for 40–60 min. Digestion was terminated by the addition of RPMI containing 10% FBS, after which the dissociated tissues were filtered through a 40-μm cell strainer (08-771-1, Thermo Fisher Scientific) then washed in PBS. The cells were resuspended in 36% Percoll solution (GE Healthcare) containing 4% 10× PBS and 60% serum-free DMEM then separated using density gradient centrifugation to remove non-immune cells. The immune cells were stimulated with a leukocyte activation cocktail (550583, BD Biosciences), incubated at 37 °C for 4–6 h in accordance with the manufacturer's instructions, and stained with a LIVE/DEAD Fixable Violet dead cell staining kit on ice for 30 min in the dark. The cells were then washed with PBS, blocked using TruStain FcX™ (anti-mouse CD16/32 antibody) then stained on ice for the cell surface epitopes CD45, CD3, and CD8 in PBS supplemented with 2% FBS for 20 min in the dark. The cells were washed again with PBS, fixed then permeabilized using a fixation/permeabilization solution kit (555028, BD Biosciences). The cells were also stained for intracellular Granzyme B, Perforin, TNF-α, and IFN-γ in permeabilization solution (555028, BD Biosciences). All samples were analyzed by flow cytometry using a Beckman CytoFLEX LX and data was analyzed by FlowJo software.

**Profile analysis of immunosuppressive population in PDAC tumors.** Patient samples were dissociated and digested, as described above. The cells were stained using a LIVE/DEAD Fixable Violet dead cell stain kit and incubated for 30 min on ice in the dark. The cells were then washed with PBS, blocked using human TruStain FcX™ (Fc receptor blocking solution), stained with antibodies for the cell surface epitopes CD45, C11b, CD11c, CD15, CD68, and HLA-DR dissolved in PBS

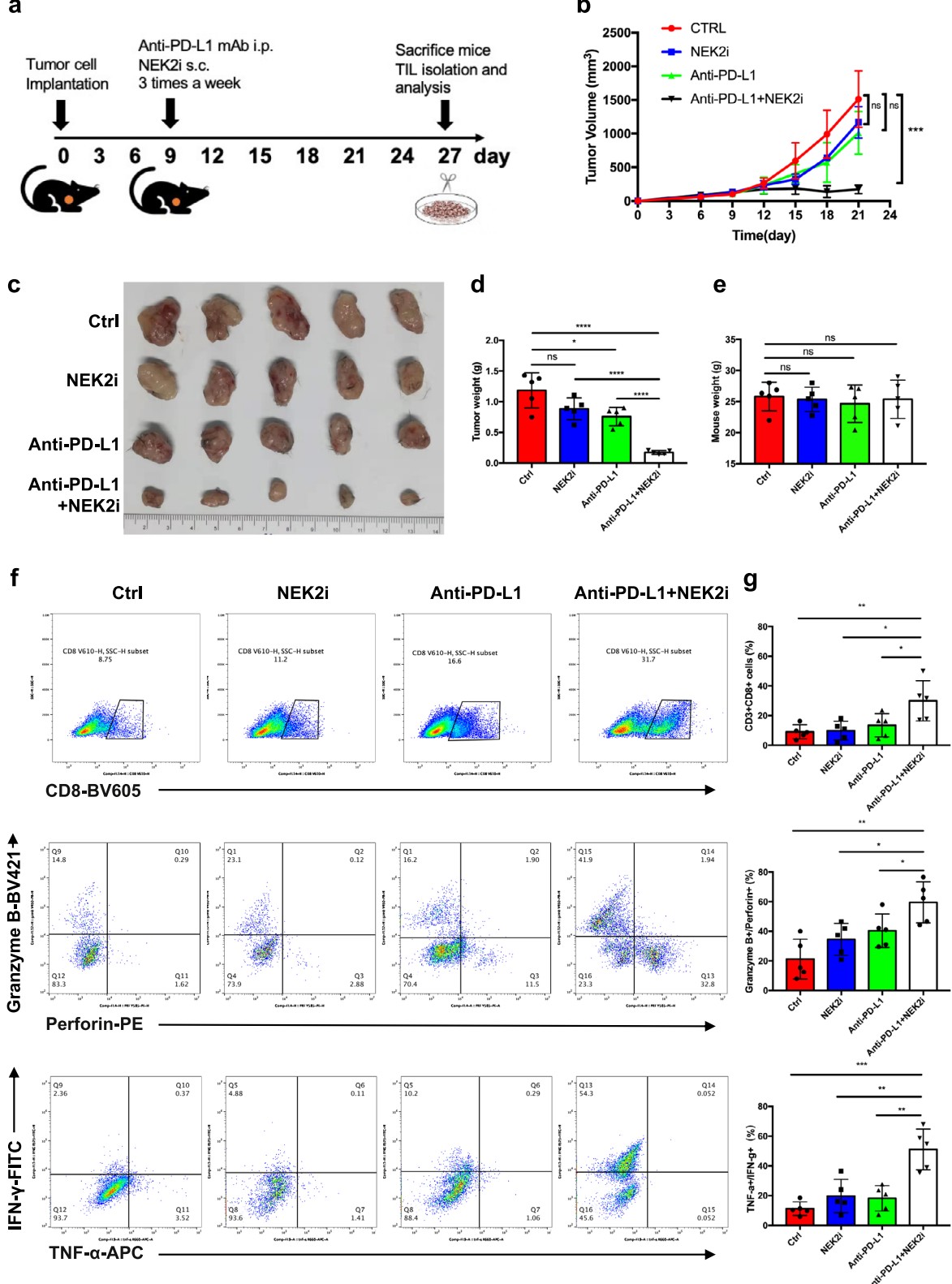

**Fig. 7 NEK2 inhibition sensitizes PD-L1-targeted pancreatic cancer immunotherapy. a** Schematic protocol of the combination of anti-PD-L1 antibody and NEK2 inhibitor therapy. **b** Tumor growth curve of mice treated with anti-PD-L1 antibody (200 μg/mouse), NEK2 inhibitor (100 μg/mouse), or their combination (n = 5). **c** Representative images displaying tumors harvested from mice bearing KPC cells treated with anti-PD-L1 antibody, NEK2 inhibitor, or their combination (n = 5). **d**, **e** Tumor weight and mouse body weight (n = 5). **f**, **g** Flow cytometric analysis and statistical results of lymphocytes that have infiltrated the tumors (n = 5). Results are presented as means ± SD from one representative experiment in **b**, **d**, **e**, and **g**. All data are representative of three independently performed experiments. *P < 0.05, **P < 0.01, ***P < 0.001 using a two-tailed t-test; ns: not significant.

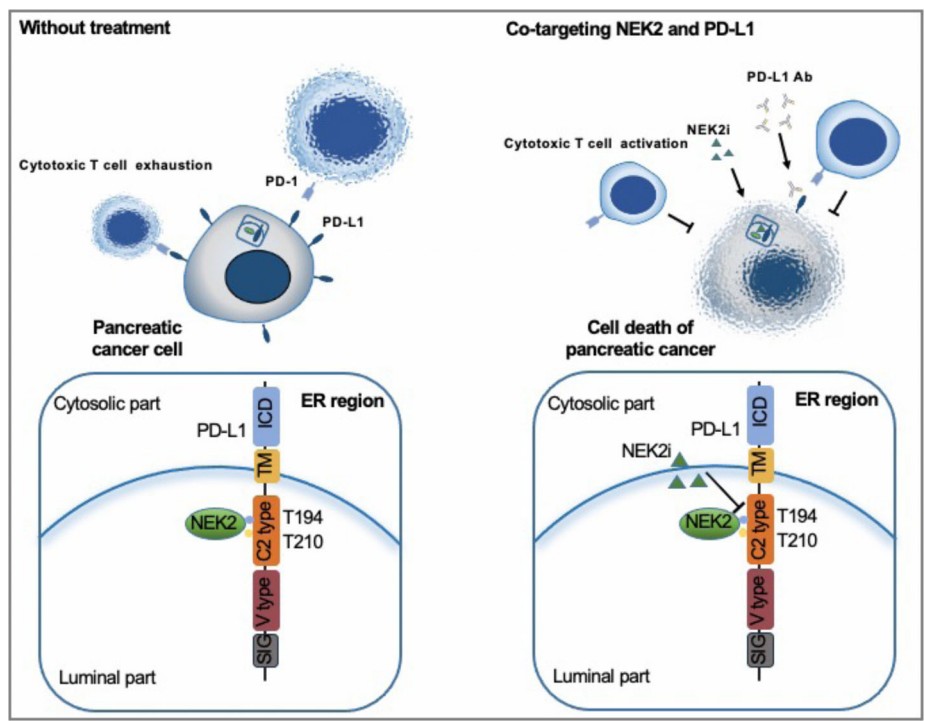

**Fig. 8 Predicted model of the NEK2–PD-L1 signaling pathway in pancreatic cancer.** A schematic model is proposed to illustrate how PD-L1 protein stability is regulated by NEK2 in pancreatic cancer. NEK2 positively regulates and interacts with PD-L1 largely through PD-L1 phosphorylation at the T194/T210 residue in ER of pancreatic cancer. Therefore, treatment with NEK2 inhibitor unexpectedly suppressed PD-L1 protein expression, largely by inhibition of PD-L1 phosphorylation that promotes its degradation.

containing 2% FBS, then incubated for 30 min on ice in the dark. The cells were washed again with PBS, fixed and permeabilized using fixation/permeabilization solution kit (555028, BD Biosciences), then stained for intracellular NEK2 (610594, BD Biosciences) in permeabilization solution (555028, BD Biosciences). The cells were washed again with permeabilization solution and stained for PE rat anti-mouse IgG1 (550083, BD Biosciences). All samples were analyzed by flow cytometry using a Beckman CytoFLEX LX. All flow cytometry data were analyzed using FlowJo software.

**Antibody generation and detection of anti-phospho-T194/T210-PD-L1 antibodies.** To generate custom antibodies against PD-L1, a 10–16-week process comprising four phases was conducted, as follows. Phase I included the design and synthesis of the T194/T210-PD-L1 peptide (1–3 weeks). Conjugated peptides were then used to immunize rabbits through 4–6 rounds of immunization in phase II (8–12 weeks). Pre-immune (5 mL) and immunized blood (100–120 mL) were collected then immunized blood precipitated using ammonia persulfate, and antibodies purified by procedures that included antigen affinity and special affinity purification. The resultant samples were suspended in PBS containing 0.03% sodium azide (phase III). The specificity of each purified antibody was verified in phase IV.

**LC-MS/MS analysis.** Mass spectrometry analysis was performed by PTM-Biolabs Co., Ltd. (Hangzhou, China). MS/MS data were processed using a Maxquant search engine (v.1.5.2.8). Tandem mass spectra were searched against the human uniprot database concatenated with a reverse decoy database. Trypsin/P was specified as the cleavage enzyme, allowing up to 4 missing cleavages. Mass tolerance for precursor ions was set to 20 ppm in the initial search and 5 ppm in the main search. Mass tolerance for fragment ions was set to 0.02 Da. Cysteine carbamidomethylation was specified as a fixed modification and acetylation and oxidation of Met were specified as variable modifications. FDR was adjusted to <1% with a minimum score for modified peptides set to >40. Proteins were classified using GO annotation into three categories: biological process, cellular compartment, and molecular function. For each category, a two-tailed Fisher's exact test was used to test for enrichment of differentially expressed proteins against all identified proteins. Corrected $p$-values <0.05 were considered statistically significant.

**Quantification and statistical analysis.** Statistical analysis was performed using SPSS (V20, IBM Corp., Armonk, NY, USA) and Prism software (GraphPad Inc, version 7.0). Data for all experiments from at least three biological replicates are presented as means ± SD. The significance of the difference between the two groups was evaluated using a Student's $t$-test, while one-way and repeated-measures analysis of variance (ANOVA) was used to compare multiple groups. Spearman's rank correlation was performed to analyze the correlation between variables. The overall difference in data at the endpoint was assessed using a Student's $t$-test to evaluate tumor growth. The difference in survival curves was analyzed using the Kaplan–Meier method and a Gehan–Breslow–Wilcoxon test. $P$ values < 0.05 were considered statistically significant.

**Reporting summary.** Further information on research design is available in the Nature Research Reporting Summary linked to this article.

## Data availability

Figure 1a, e–h, Supplementary Figs. 1a, b, 2a, b, 4a, b, 6a and b were generated from publicly available databases from TCGA, ICGC, and GTEx by GEPIA2 (http://gepia2.cancer-pku.cn), TISIDB (http://cis.hku.hk/TISIDB), Kaplan–Meier Plotter (http://kmplot.com/analysis), and cBioPortal for cancer genomics (http://www.cbioportal.org). The proteomic data have been deposited in PRIDE database under the accession code PXD026877. The remaining data are available within the Article, Supplementary Information or Source Data file. Source data are provided with this paper.

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

## Acknowledgements

This study was supported by the National Key Research and Development Program (2019YFC1316000 to T.L.), National Natural Science Foundation of China (81830089 to T.L.; 81871925 and 82071867 to X.B.; 31970696 and 81502975 to X.H.), Key Research and Development Program of Zhejiang Province (2020C03117 to X.B.), and China Postdoctoral Science Foundation (2016T90413 and 2015M581693 to X.H.). No aspect of the study, including study design, the collection, analysis, and interpretation of the data, and writing of the report, was influenced by the sponsoring foundations. We thank Mr. Minghua Sun, Mr. Jianfeng Wang, and Mr. Lei Ni for their technical support during sample processing and assistance with mouse experiments. The author X.H. would like to express deepest thanks to Dr. Guido Kroemer (Gustave Roussy Comprehensive Cancer Institute), Dr. Wei Xie (Southeast University), and Dr. Mian Wu (University of Science and Technology of China) for training in their labs.

## Author contributions

T.-B.L., X.-L.B., and X.H. supervised the study. X.H. conceived the project and designed the experiments; X.-Z.Z. performed the majority of experiments and acquired the data; X.H. conducted data analyses and interpretation; J.X., E.-L.L., M.-Y.L., T.-Y.T., G.Z., C.-X.G., X.-Y.Z., W.C., and D.K.Y. contributed to technical assistance. X.H. and X.-Z.Z. wrote the draft; X.H., X.-Z.Z., and G.Z. revised the manuscript; all other authors discussed and commented on the manuscript. X.H. and X.-Z.Z. contributed equally to this work; T.-B.L., X.-L.B., and X.H. share the senior authorship of this study. All authors read and approved the final manuscript.

## Competing interests

The authors declare no competing interests.
