## [Peer Review File · Nature Communications]

REVIEWER COMMENTS

Reviewer #1 (Remarks to the Author): with expertise in pancreatic cancer and immunology/immunotherapy

Zhang et al provide a novel finding of NEK2 in regulating expression of PD-L1 expression in pancreatic cancer and provide evidence that targeting NEK2 can improve the efficacy in pre-clinical animal models. This manuscript is comprehensive in utilizing patient data/tissue and several different animal models of pancreatic cancer. Additionally, they very elegantly test the mechanism of NEK2 phosphorylation of PD-L1 by modifying phosphorylation sites and assessing NEK2 binding and activity. Findings in this manuscript are indeed novel, however several areas require revisions to improve the quality. My major concerns and comments are listed below:

Major concerns:

Overall, grammar and flow of the writing in this manuscript needs to be carefully reviewed to improve the quality for reading this work.

Figure legends: All of the figure legends need severe editing and revisions. Legends are too abbreviated and lack essential details. Figure legends need to describe the experiment that was completed and not the result/conclusion from that figure. Example Line 799 which states "(D-K) NEK2 controls the protein stability of PD-L1 in a kinase-dependent manner." This goes in the results section. Authors should be describing what was done in experiments D-K. Author glosses over several parts with one sentence. This needs to be corrected throughout all the legends and needs to be greatly improved.

Line 26: Abstract: Authors do not clearly state a central hypothesis.

Line 60: Introduction – providing some recent publications of clinical trials where checkpoint immunotherapy have failed in PDAC would improve the introduction and would provide a better context for the reader on the importance of the work presented in this manuscript.

Line 61: 7% 5- year survival. Is this world-wide, US...please clarify.

Lines 103-105: Is NEK2 expressed by macrophages, DC, MDSC? These suppressive populations are highly elevated in PDAC patients and greatly influence the poor efficacy of checkpoint therapy. Could NEK2 expression by MDSC or other suppressive immune populations contribute to greater PD-L1 on these immune populations? PD-L1 is not only expressed by tumor cells, but these immune populations. Greater context here or in the discussion should be provided.

Line 158: Details on phenotype of the cell line should be provided in the results. Yes, it is provided in methods, but this should be reiterated here as well.

Line 180: What is the "homonymous assay" described in the results section? This is very unclear and is never described again throughout the paper or methods. Please clarify or remove this text.

Line 185: The NEK2 inhibitor is used throughout this paper, however details regarding this inhibitor are severely lacking. Commercial or proprietary reagent? Dose known to inhibit NEK2 in vitro and in vivo? References to prior use of this inhibitor. Greater details both in results and methods are required.

Line 209: What is "T" and "N" in figure? No labels in the methods. Is this normal and tumor tissue? If so, where was normal pancreatic tissue acquired to make these lysates? Lysates from tumor tissue are also comprised of stromal cells...how did you account for expression of NEK2 by stromal populations in these lysates?

Line 239: Statistics/densitometry or even mention of whether these findings were repeated are absent. No mention in the text or methods. Some densitometry and evidence that findings were repeated are required. Do not need to quantify every western blot as that would probably be unreasonable, however the authors should provide some evidence where experiments were

repeated in triplicate and densitometry was performed. Rigor and reproducibility are very important. Figure 5 would be a good example of where this should be provided.

Line 25: Checkpoint immunotherapy alone can inhibit tumor growth in subcutaneous models of PDAC, however single agent therapy in patients is ineffective. However, single agent checkpoint therapy does not work in GEM models. Does NEK2 + PD-L1 combination hold up in a GEM model of PDAC? Or an orthotopic model where stromal involvement is more present than subcutaneous models used in this paper.

Minor:

Line 145: should read, "Unexpectedly, NEK2 expression was..."

Lines 172-175: These sentences are very confusing to read and could be edited for better clarity.

Line 319: Should cite references where NEK2 can regulate phosphorylation of other proteins.

Line 571: Unclear when reading what experiments are subcut and which ones are orthotopic. I didn't even realize orthotopic was completed until I read the methods as no mention is found in the results or figure legends. Please clarify throughout in the results and figure legends.

Reviewer #2 (Remarks to the Author): with expertise in NEK2

This is an interesting paper which includes novel NEK2 functions in immune resistance to immune checkpoint blockade. However, there are some important questions before this manuscript is accepted for a formal publication.

1. What is the NEK2 inhibitor? What is the inhibitor name? where did the authors receive them? No any data were described about this NEK2 inhibitor in this paper. Does this inhibitor degrade NEK2 protein or inhibit its kinase function?

2. In line with the above issue, no data support whether the NEK2 inhibitor is functional inhibition of NEK2 in the in vivo mouse model (Figure 7).

3. If NEK2 phosphates PD-L1 resulting in its stabilization, the author also need to mutate NEK2 (NEK2-dead: K37R inactive mutant) to repeat and confirm the results shown in the Figure 6.

4. It is not clear why the author chose to study NEK2 in pancreatic cancer. The rationale is not clear even they showed NEK2 expression is negatively correlated with patient outcome.

Reviewer #3 (Remarks to the Author): with expertise in PD-L1 - mechanisms of regulation

The reviewer has three major concerns about this paper:

1) The expression of PD-L1 seems to positive in most of the samples shown in the paper, but it is generally believed that PD-L1 is only expressed in a smaller subset of tumors. For instance, the ProteinAtlas database shows that PD-L1 is basically negative in pancreatic cancer. Thus, the correlation between NEK2 and PD-L1 does not seem to be convincing enough.

2) The relationship between the roles of NEK2 and GSK3B should be clarified, plus the experiments showing the relative importance of both regulators on PD-L1 expression.

3) NEK2 has been established as an anti-cancer target, and it is unclear if NEK2 may regulate anti-tumor immunity through any other signaling pathways. For example, previous studies have implicated Nek2 in early B cell development and germinal center formation.

<https://www.ncbi.nlm.nih.gov/pmc/articles/PMC4251609>

Reviewer #1 (Remarks to the Author): with expertise in pancreatic cancer and immunology/immunotherapy

Zhang et al provide a novel finding of NEK2 in regulating expression of PD-L1 expression in pancreatic cancer and provide evidence that targeting NEK2 can improve the efficacy in pre-clinical animal models. This manuscript is comprehensive in utilizing patient data/tissue and several different animal models of pancreatic cancer. Additionally, they very elegantly test the mechanism of NEK2 phosphorylation of PD-L1 by modifying phosphorylation sites and assessing NEK2 binding and activity. Findings in this manuscript are indeed novel, however several areas require revisions to improve the quality. My major concerns and comments are listed below:

Response: We thank you for your encouraging comments. All your concerns are addressed in a point-by-point manner, as follows.

Major concerns:

Overall, grammar and flow of the writing in this manuscript needs to be carefully reviewed to improve the quality for reading this work.

Response: Thank you for your critical comments. We have now rewritten almost every part of the manuscript, and writing quality (including but not limited to grammar and flow) has been greatly improved for better reading and understandability.

Figure legends: All of the figure legends need severe editing and revisions. Legends are too abbreviated and lack essential details. Figure legends need to describe the experiment that was completed and not the result/conclusion from that figure. Example

Line 799 which states “(D-K) NEK2 controls the protein stability of PD-L1 in a kinase-dependent manner.” This goes in the results section. Authors should be describing what was done in experiments D-K. Author glosses over several parts with one sentence. This needs to be corrected throughout all the legends and needs to be greatly improved.

Response: Thank you for your critical comments and constructive suggestions. We have now rewritten all Figure legends, which we believe are a great improvement, as requested.

Line 26: Abstract: Authors do not clearly state a central hypothesis.

Response: Thank you for your critical comment. We have now rewritten the whole Abstract for clarity of the purpose of the paper.

Line 60: Introduction – providing some recent publications of clinical trials where checkpoint immunotherapy have failed in PDAC would improve the introduction and would provide a better context for the reader on the importance of the work presented in this manuscript.

Response: Thank you for your constructive suggestions. As suggested, we have now added several recent publications of clinical trials where checkpoint immunotherapy have failed in PDAC, providing a clinical context so that the reader can better understand the potential significance of the findings presented in this study.

Line 61: 7% 5- year survival. Is this world-wide, US...please clarify.

Response: Thank you for your constructive suggestions. We have now rewritten this sentence to highlight that this is a global problem.

Lines 103-105: Is NEK2 expressed by macrophages, DC, MDSC? These suppressive populations are highly elevated in PDAC patients and greatly influence the poor efficacy of checkpoint therapy. Could NEK2 expression by MDSC or other suppressive immune populations contribute to greater PD-L1 on these immune populations? PD-L1 is not only expressed by tumor cells, but these immune populations. Greater context here or in the discussion should be provided.

Response: Thank you for your constructive suggestions. As suggested, we have now performed additional experiments by FACS to test the expression of NEK2 in a number of immunosuppressive populations (such as macrophage, DC, and MDSC) in comparison with that in tumors. Intriguingly, as you have supposed, NEK2 expression has also been observed in macrophages, DCs, and MDSCs. However, the proportion of NEK2 positive cells in such suppressive immune cells are significantly lower than that in tumor cells. Not only that, the total number of all immune cells is in fact considerably lower than the number of tumor cells in pancreatic cancer tissues. Therefore, NEK2 may also regulate the expression of PD-L1 in immunosuppressive populations. We have therefore focused on its regulatory effect on tumor cells in this study. We have now supplemented the observation described above across the manuscript.

Line 158: Details on phenotype of the cell line should be provided in the results. Yes, it is provided in methods, but this should be reiterated here as well.

Response: We agree that we have not provided sufficient information in the Results in this regard. We have now concisely reiterated the detailed phenotype of the cell lines as follows: “A KPC-NEK2 knockdown (KD) pancreatic cancer cell line, stably transfected with Nek2 double nickase plasmids, was generated to determine whether NEK2 is associated with anti-tumor immune response.”.

Line 180: What is the “homonymous assay” described in the results section? This is very unclear and is never described again throughout the paper or methods. Please clarify or remove this text.

Response: By “homonymous assay” we meant “T cell-mediated tumor cell-killing assay”, which we have mentioned above. We have now replaced the relevant sentence and further clarified the details of “T cell-mediated tumor cell-killing” in the Methods.

Line 185: The NEK2 inhibitor is used throughout this paper, however details regarding this inhibitor are severely lacking. Commercial or proprietary reagent? Dose known to inhibit NEK2 *in vitro* and *in vivo*? References to prior use of this inhibitor. Greater details both in results and methods are required.

Response: We agree that there was a lack of details of the NEK2 inhibitor used in this paper. It is a commercial reagent named NCL 00017509 (Cat. No. 5150, Tocris Bioscience), which shows potent and reversible inhibitory effects on NEK2, and is cell-permeable and active *in vivo*. We have now added more detailed information about the NEK2 inhibitor in the Methods section, including but not limited to, the source, catalog

no., commercial name, and chemical composition, and have also cited relevant references. Moreover, we have now supplemented, in the Results section and Figure legends, the dose and duration of treatment with the NEK2 inhibitor *in vitro* and *in vivo*.

Line 209: What is “T” and “N” in figure? No labels in the methods. Is this normal and tumor tissue? If so, where was normal pancreatic tissue acquired to make these lysates? Lysates from tumor tissue are also comprised of stromal cells...how did you account for expression of NEK2 by stromal populations in these lysates?

Response: Thank you for your critical comments. “T” refers to “pancreatic tumor tissue” and “N” indicates “normal pancreatic tissue”. We have now supplemented these abbreviations in the Figure legends, as appropriate. Moreover, the collected pancreatic tumor tissue and para-cancerous normal tissue have now been individually labeled in a representative images. We have also uploaded this image as a Supplementary file. In addition, we agree that lysates from pancreatic tumor tissue consist of stromal cell constituents, and thus the analysis of NEK2 expression between “N” and “T” using Western blotting is relatively preliminary. However, the following assays (including IHC staining and FACS) further confirm higher expression levels of NEK2 in pancreatic tumors in comparison with the normal control. We have now highlighted the above information across the manuscript.

Line 239: Statistics/densitometry or even mention of whether these findings were repeated are absent. No mention in the text or methods. Some densitometry and evidence that findings were repeated are required. Do not need to quantify every western blot as that would probably be unreasonable, however the authors should provide some evidence where experiments were repeated in triplicate and densitometry was performed. Rigor and reproducibility are very important. Figure 5 would be a good example of where this should be provided.

Response: Thank you for your critical comments and constructive suggestions. We have now quantified all important blotting bands by densitometry, and added an analysis of the statistics under the corresponding images in the Figures. In addition, all experiments, except LC-MS proteomics quantitative analysis, were repeated at least three times independently. We have also supplemented this information in the relevant Figure legends.

Line 25: Checkpoint immunotherapy alone can inhibit tumor growth in subcutaneous models of PDAC, however single agent therapy in patients is ineffective. However,

single agent checkpoint therapy does not work in GEM models. Does NEK2 + PD-L1 combination hold up in a GEM model of PDAC? Or an orthotopic model where stromal involvement is more present than subcutaneous models used in this paper.

Response: Thank you for your constructive suggestions. We have now re-performed combination therapy co-targeting NEK2 and PD-L1 using an orthotopic mouse model, and similar synergistic effects of suppression of PDAC tumor growth were observed.

Minor:

Line 145: should read, “Unexpectedly, NEK2 expression was...”

Response: We have now rephrased this sentence as instructed, highlighting correlation at the level of expression.

Lines 172-175: These sentences are very confusing to read and could be edited for better clarity.

Response: We have rephrased these sentences for better clarity, as instructed.

Line 319: Should cite references where NEK2 can regulate phosphorylation of other proteins.

Response: Thank you for your constructive suggestions. We have now cited several representative references highlighting the kinase function of NEK2, including but not limited to, its phosphorylation of p53 and GAS2L1.

Line 571: Unclear when reading what experiments are subcut and which ones are orthotopic. I didn't even realize orthotopic was completed until I read the methods as no mention is found in the results or figure legends. Please clarify throughout in the results and figure legends.

Response: We have revised the schematic diagram and added a concise description of both orthotopic and subcutaneous mouse models, also clarifying the relevant content throughout the Results and Figure legends, as instructed.

Reviewer #2 (Remarks to the Author): with expertise in NEK2

This is an interesting paper which includes novel NEK2 functions in immune resistance to immune checkpoint blockade. However, there are some important questions before this manuscript is accepted for a formal publication.

Response: Thank you for your encouraging comments. All your concerns are addressed in a point-by-point manner, as follows.

1. What is the NEK2 inhibitor? What is the inhibitor name? where did the authors receive them? No any data were described about this NEK2 inhibitor in this paper. Does this inhibitor degrade NEK2 protein or inhibit its kinase function?

Response: We agree that the details of the NEK2 inhibitor were lacking in this paper. It is a commercial reagent called NCL 00017509 (Cat. No. 5150, Tocris Bioscience), which shows potent and reversible inhibitory effects on NEK2, is cell permeable, and active *in vivo*. We have now added more detailed information about the NEK2 inhibitor in the Methods, including but not limited to, its source, catalog no., commercial name, and chemical composition, and have also cited relevant references. Moreover, we have now supplemented, in the Results and Figure legends, dose and duration when using it in treatments with the NEK2 inhibitor both *in vitro* and *in vivo*. Furthermore, the experimental outcomes of this study confirm that NCL 00017509 inhibits the kinase function of NEK2, rather than mediating protein degradation.

2. In line with the above issue, no data support whether the NEK2 inhibitor is functional inhibition of NEK2 in the *in vivo* mouse model (Figure 7).

We have measured both the expression level and phosphorylation state of NEK2 in tumors that have received combinatorial therapy (including NEK2 inhibitor alone or together with an anti-PD-L1 antibody) in an orthotopic mouse model, and confirmed that NCL 00017509 inhibits the kinase function of NEK2, rather than mediating protein degradation, *in vivo*.

3. If NEK2 phosphates PD-L1 resulting in its stabilization, the author also need to mutate NEK2 (NEK2-dead: K37R inactive mutant) to repeat and confirm the results shown in the Figure 6.

Response: Thank you for your constructive suggestions. As you suggest, we have now used an inactive mutant of NEK2 (K37R, a kinase-dead mutant) in NEK2-depleted KPC cells, and found the mutation indeed caused NEK2 to have loss of function in the regulation of expression, half-life, and ubiquitination of PD-L1, confirming that NEK2 phosphorylates PD-L1, resulting in its stabilization.

4. It is not clear why the author chose to study NEK2 in pancreatic cancer. The rationale is not clear even they showed NEK2 expression is negatively correlated with patient outcome.

Response: Our research group focuses on pancreatic diseases, especially pancreatic cancer, in which PD-1/PD-L1 inhibitors show low efficacy and effectiveness. Increasing evidence suggests that post-translational modification of PD-L1 largely determines its targeted immunotherapy. At the beginning of this study, we found that there are two conserved modification motifs of NEK family on PD-L1. Furthermore,

using TCGA datasets, we investigated the differential expression and prognostic relevance of all members of the NEK family in pancreatic cancer, finding potential clinical significance of NEK2 in such disease. In addition, NEK2 participates in the regulation of the cell cycle and mitosis, while regulators of the cell cycle (including but not limited to CDK4/5/6) are closely related to the expression level of PD-L1 and its mediation of cancer immune resistance. Therefore, we chose to study NEK2 in pancreatic cancer immunity. We have now highlighted the rationale of this study in the Introduction and supplemented the whole manuscript with additional information.

Reviewer #3 (Remarks to the Author): with expertise in PD-L1 - mechanisms of regulation

The reviewer has three major concerns about this paper:

Response: We thank the reviewer for evaluating the manuscript. All the reviewer's concerns have been addressed in a point-by-point manner as follows.

1) The expression of PD-L1 seems to positive in most of the samples shown in the paper, but it is generally believed that PD-L1 is only expressed in a smaller subset of tumors. For instance, the ProteinAtlas database shows that PD-L1 is basically negative in pancreatic cancer. Thus, the correlation between NEK2 and PD-L1 does not seem to be convincing enough.

Response: Thank you for your critical comments. Firstly, we disagree that “The expression of PD-L1 seems positive in most of the samples shown in the paper.”. Actually, we performed a pancreatic cancer tissue microarray (n=156) and found that approximately one half of the cases showed little or no expression of PD-L1, as well as relatively low expression of NEK2, indicating a positive correlation between PD-L1 and NEK2 expression in pancreatic cancer. Furthermore, although the Protein Atlas database shows that PD-L1 is essentially negative in pancreatic cancer, the number of patient samples detected in the Protein Atlas database is rather fewer than in our study using a tissue microarray. In addition, we need to mention that the antibodies used to detect PD-L1 expression in the Protein Atlas and in our study are also different, which may be additional important reason for such a difference. Although both antibodies have been validated as specific against PD-L1, the antibody used in our study has

received many citations. We have added the information above throughout the manuscript.

2) The relationship between the roles of NEK2 and GSK3B should be clarified, plus the experiments showing the relative importance of both regulators on PD-L1 expression.

Response: Thank you for your constructive suggestions. As suggested, we have now performed additional experiments to investigate the relative importance of NEK2 and GSK3 β on PD-L1 expression in the context of our study (in other words, in pancreatic cancer). Interestingly, we found that NEK2 inhibition led to inactivation of GSK3 β (decreased p-GSK3 β) and downregulation of PD-L1 (consistent with our previous results in this study) in a dose-dependent manner. However, the effects of GSK3 β inhibitor on NEK2 and PD-L1 expression are not stable enough for conclusive conclusions to be made. Overall, the results suggest that NEK2, rather than GSK3 β , plays a relatively dominant role in the regulation of PD-L1 expression in pancreatic cancer. We have now supplemented the manuscript with these data.

3) NEK2 has been established as an anti-cancer target, and it is unclear if NEK2 may regulate anti-tumor immunity through any other signaling pathways. For example, previous studies have implicated Nek2 in early B cell development and germinal center formation. <https://www.ncbi.nlm.nih.gov/pmc/articles/PMC4251609>

Response: Thank you for your insightful comments. We have now rechecked the previous data for NEK2 overexpression-induced differential expression of proteins, in

which we identified PD-L1 as a potential target of NEK2 in pancreatic cancer. Gene Ontology enrichment analysis revealed that NEK2 is closely related to multiple aspects of immune regulation, including but not limited to, cytokine and cytokine receptor binding, interleukin production, leukocyte proliferation, migration, and activation, in addition to phagocytosis and endocytosis. We also need to mention that NEK2 expression has also been observed in suppressive immune cells (such as macrophages, DCs, and MDSCs) in our study, suggesting that NEK2 may be involved in tumor immune resistance through regulation of these immunosuppressive populations. We have now supplemented the manuscript with these aspects.

REVIEWER COMMENTS

Reviewer #1 (Remarks to the Author):

The authors have thoroughly addressed all of my concerns.

Reviewer #4 (Remarks to the Author): to replace original reviewers #2-3

My review consists with two-part.

1. Verification of responses to previous comments from reviewers #2 and #3.
Due to the unavailability of reviewers #2 and #3, I (reviewer #4) have been invited additionally. Therefore, at first, I examined the author's responses to previous comments from reviewers #2 and #3.

2. Reviewers #4's comments.

After this revision, the manuscript is highly improved, however, unfortunately, a very important point to support the author's hypothesis is missing. I additionally commented to make up for this.

1. Verification of responses to previous comments from reviewers #2 and #3.

Reviewer #2 (Remarks to the Author): with expertise in NEK2

This is an interesting paper which includes novel NEK2 functions in immune resistance to immune checkpoint blockade. However, there are some important questions before this manuscript is accepted for a formal publication.

Response: Thank you for your encouraging comments. All your concerns are addressed in a point-by-point manner, as follows.

1. What is the NEK2 inhibitor? What is the inhibitor name? where did the authors receive them? No any data were described about this NEK2 inhibitor in this paper. Does this inhibitor degrade NEK2 protein or inhibit its kinase function?

Response: We agree that the details of the NEK2 inhibitor were lacking in this paper. It is a commercial reagent called NCL 00017509 (Cat. No. 5150, Tocris Bioscience), which shows potent and reversible inhibitory effects on NEK2, is cell permeable, and active in vivo. We have now added more detailed information about the NEK2 inhibitor in the Methods, including but not limited to, its source, catalog no., commercial name, and chemical composition, and have also cited relevant references. Moreover, we have now supplemented, in the Results and Figure legends, dose and duration when using it in treatments with the NEK2 inhibitor both in vitro and in vivo. Furthermore, the experimental outcomes of this study confirm that NCL 00017509 inhibits the kinase function of NEK2, rather than mediating protein degradation.

Reviewer#4's comment: The author provides enough information to address for the comment

2. In line with the above issue, no data support whether the NEK2 inhibitor is functional inhibition of NEK2 in the in vivo mouse model (Figure 7).

We have measured both the expression level and phosphorylation state of NEK2 in tumors that have received combinatorial therapy (including NEK2 inhibitor alone or together with an anti-PD-L1 antibody) in an orthotopic mouse model, and confirmed that NCL 00017509 inhibits the kinase function of NEK2, rather than mediating protein degradation, in vivo.

Reviewer#4's comment: Great. It was addressed experimentally.

3. If NEK2 phosphates PD-L1 resulting in its stabilization, the author also need to mutate NEK2 (NEK2-dead: K37R inactive mutant) to repeat and confirm the results shown in the Figure 6.

Response: Thank you for your constructive suggestions. As you suggest, we have now used an

inactive mutant of NEK2 (K37R, a kinase-dead mutant) in NEK2-depleted KPC cells, and found the mutation indeed caused NEK2 to have loss of function in the regulation of expression, half-life, and ubiquitination of PD-L1, confirming that NEK2 phosphorylates PD-L1, resulting in its stabilization.

Reviewer#4's comment: Great. It was addressed experimentally.

4. It is not clear why the author chose to study NEK2 in pancreatic cancer. The rationale is not clear even they showed NEK2 expression is negatively correlated with patient outcome.

Response: Our research group focuses on pancreatic diseases, especially pancreatic cancer, in which PD-1/PD-L1 inhibitors show low efficacy and effectiveness. Increasing evidence suggests that post-translational modification of PD-L1 largely determines its targeted immunotherapy. At the beginning of this study, we found that there are two conserved modification motifs of NEK family on PD-L1. Furthermore, using TCGA datasets, we investigated the differential expression and prognostic relevance of all members of the NEK family in pancreatic cancer, finding potential clinical significance of NEK2 in such disease. In addition, NEK2 participates in the regulation of the cell cycle and mitosis, while regulators of the cell cycle (including but not limited to CDK4/5/6) are closely related to the expression level of PD-L1 and its mediation of cancer immune resistance. Therefore, we chose to study NEK2 in pancreatic cancer immunity. We have now highlighted the rationale of this study in the Introduction and supplemented the whole manuscript with additional information.

Reviewer#4's comment: In the current version, there seems to be no problem with the legitimacy of the study.

Reviewer #3 (Remarks to the Author): with expertise in PD-L1 - mechanisms of regulation
The reviewer has three major concerns about this paper:

Response: We thank the reviewer for evaluating the manuscript. All the reviewer's concerns have been addressed in a point-by-point manner as follows.

1) The expression of PD-L1 seems to be positive in most of the samples shown in the paper, but it is generally believed that PD-L1 is only expressed in a smaller subset of tumors. For instance, the ProteinAtlas database shows that PD-L1 is basically negative in pancreatic cancer. Thus, the correlation between NEK2 and PD-L1 does not seem to be convincing enough.

Response: Thank you for your critical comments. Firstly, we disagree that "The expression of PD-L1 seems positive in most of the samples shown in the paper.". Actually, we performed a pancreatic cancer tissue microarray (n=156) and found that approximately one half of the cases showed little or no expression of PD-L1, as well as relatively low expression of NEK2, indicating a positive correlation between PD-L1 and NEK2 expression in pancreatic cancer. Furthermore, although the Protein Atlas database shows that PD-L1 is essentially negative in pancreatic cancer, the number of patient samples detected in the Protein Atlas database is rather fewer than in our study using a tissue microarray. In addition, we need to mention that the antibodies used to detect PD-L1 expression in the Protein Atlas and in our study are also different, which may be an additional important reason for such a difference. Although both antibodies have been validated as specific against PD-L1, the antibody used in our study has received many citations. We have added the information above throughout the manuscript.

Reviewer#4's comment: The author's answer is reasonable and I have no argument about the positive correlation between PD-L1 and NEK2 in the current version.

2) The relationship between the roles of NEK2 and GSK3B should be clarified, plus the experiments showing the relative importance of both regulators on PD-L1 expression.

Response: Thank you for your constructive suggestions. As suggested, we have now performed additional experiments to investigate the relative importance of NEK2 and GSK3 β on PD-L1 expression in the context of our study (in other words, in pancreatic cancer). Interestingly, we

found that NEK2 inhibition led to inactivation of GSK3 β (decreased p-GSK3 β) and downregulation of PD-L1 (consistent with our previous results in this study) in a dose-dependent manner. However, the effects of GSK3 β inhibitor on NEK2 and PD-L1 expression are not stable enough for conclusive conclusions to be made. Overall, the results suggest that NEK2, rather than GSK3 β , plays a relatively dominant role in the regulation of PD-L1 expression in pancreatic cancer. We have now supplemented the manuscript with these data.

Reviewer#4's comment: GSK3 β cannot bind to normally glycosylated-PD-L1 and it only phosphorylates T180 and S184 of non-glycosylated-PD-L1. This phosphorylation induces degradation of non-glycosylated PD-L1. This event is restricted to non-glycosylated PD-L1, not glycosylated PD-L1. (Nature Communications. 2016;7:12632. Supplementary Figure5). Logically, GSK3b inhibitor should increase the level of non-glycosylated PD-L1. Additional data showed only the level of glycosylated PD-L1, and this data confuses. I think this data is not essential to support the hypothesis of this manuscript. Alternatively, if phosphorylation by NEK2 can affect the glycosylation of PD-L1, it is advisable to check the binding of GSK3b to PD-L1.

3) NEK2 has been established as an anti-cancer target, and it is unclear if NEK2 may regulate anti-tumor immunity through any other signaling pathways. For example, previous studies have implicated Nek2 in early B cell development and germinal center formation. <https://www.ncbi.nlm.nih.gov/pmc/articles/PMC4251609>

Response: Thank you for your insightful comments. We have now rechecked the previous data for NEK2 overexpression-induced differential expression of proteins, in which we identified PD-L1 as a potential target of NEK2 in pancreatic cancer. Gene Ontology enrichment analysis revealed that NEK2 is closely related to multiple aspects of immune regulation, including but not limited to, cytokine and cytokine receptor binding, interleukin production, leukocyte proliferation, migration, and activation, in addition to phagocytosis and endocytosis. We also need to mention that NEK2 expression has also been observed in suppressive immune cells (such as macrophages, DCs, and MDSCs) in our study, suggesting that NEK2 may be involved in tumor immune resistance through regulation of these immunosuppressive populations. We have now supplemented the manuscript with these aspects.

Reviewer#4's comment: I think authors do not need to be defensive about this comment. Of course, NEK2 is probably multifunctional and NEK2i could affect other parts related to the anti-tumor effect. Reviewer #3 may want to discuss this point in the extended view beyond the PD-L1 regulation mechanism. Such an extended discussion would give deep insight into this manuscript.

2. Reviewers #4's comments.

General comment

In this manuscript, authors identify the new regulatory mechanisms of PD-L1 in PDAC, and based on this finding, they suggest a potential combination to enhance the efficacy of ICB against PDAC. There seems to be no problem with the legitimacy of the study and the correlation between NEK2 and PD-L1 is clearly shown through multiple approaches including database, TMA, animal model, and in vitro assay. The effect of NEK2 on PD-L1 and anti-tumor immunity is recognized. However, there is a fundamental problem with their hypothesis, which can be controversial. To resolve this issue, I think we need additional experimental evidence.

Major comment

Because T194/T210 is located in the extracellular domain of PD-L1, T194/T210 is only exposed inside of ER and Golgi lumen during transport into the plasma membrane. It is probably rare that T194/T210 is exposed to the cytosolic part. Therefore, Author should show evidence NEK2 can directly bind to PD-L1 and phosphorylates T194/T210 of PD-L1 in ER and/or Golgi.

1. Authors need to check the localization of NEK2 in ER or/and Golgi. Trypsinization assay with ER fraction will be available to check this issue. The Journal of Clinical Investigation. 2019 Jul 15;129(8):3324-3338, Molecular Cell. 2018;71(4):606-620

2. Although they show the binding between PD-L1 and NEK2 with endo-IP, actually, This analysis

cannot exclude the involvement of other factors. Traditional GST(or His)-full down assay with purified PD-L1 and NEK2 will be suitable and Duo-link assay with antibodies against PD-L1 and NEK2 will support real binding in the cell, not artificial tube.

3. All current data related to phosphorylation is indirect evidence to show NEK2 signaling can phosphorylate PD-L1. Other factors in NEK2 signaling may phosphorylate T194/T210. Authors have already acquired specific Abs against T194/T210 and active NEK2 (14-545M, sigma MERK) and PD-L1 are commercially available, and therefore, why not perform in vitro kinase assay. This result will be critical evidence to clearly show NEK2 directly phosphorylate T194/T210 of PD-L1.

4. If authors acquire these data, it is better to gain all data related to binding and phosphorylation and reorganize them in another figure.

I think this issue had to be indicated in the first revision. However, with these evidences, it will be a complete manuscript and there will be no boring argument related to phosphorylation of an extracellular domain in the respect of the transmembrane protein-topology.

Minor comment

In the main text, there is no description of Supplementary Fig.8.

REVIEWER COMMENTS AND POINT-BY-POINT RESPONSE

Reviewer #1 (Remarks to the Author):

The authors have thoroughly addressed all of my concerns.

Response (2nd Round): Thank you for your encouraging comments.

Reviewer #4 (Remarks to the Author): to replace original reviewers #2-3

My review consists with two-part.

1. Verification of responses to previous comments from reviewers #2 and #3. Due to the unavailability of reviewers #2 and #3, I (reviewer #4) have been invited additionally. Therefore, at first, I examined the author's responses to previous comments from reviewers #2 and #3.

2. Reviewers #4's comments. After this revision, the manuscript is highly improved, however, unfortunately, a very important point to support the author's hypothesis is missing. I additionally commented to make up for this.

Response (2nd Round): Thank you very much for your efforts to improve this study.

1. Verification of responses to previous comments from reviewers #2 and #3.

Reviewer #2 (Remarks to the Author): with expertise in NEK2

This is an interesting paper which includes novel NEK2 functions in immune resistance to immune checkpoint blockade. However, there are some important questions before this manuscript is accepted for a formal publication.

Response (1st Round): Thank you for your encouraging comments. All your concerns are addressed in a point-by-point manner, as follows.

1. What is the NEK2 inhibitor? What is the inhibitor name? where did the authors receive them? No any data were described about this NEK2 inhibitor in this paper. Does this inhibitor degrade NEK2 protein or inhibit its kinase function?

Response (1st Round): We agree that the details of the NEK2 inhibitor were lacking in this paper. It is a commercial reagent called NCL 00017509 (Cat. No. 5150, Tocris Bioscience), which shows potent and reversible inhibitory effects on NEK2, is cell permeable, and active in vivo. We have now added more detailed information about the NEK2 inhibitor in the Methods, including but not limited to, its source, catalog no., commercial name, and chemical composition, and have also cited relevant references. Moreover, we have now supplemented, in the Results and Figure legends, dose and duration when using it in treatments with the NEK2 inhibitor both in vitro and in vivo. Furthermore, the experimental outcomes of this study confirm that NCL 00017509 inhibits the kinase function of NEK2, rather than mediating protein degradation.

Reviewer#4's comment: The author provides enough information to address for the comment.

Response (2nd Round): Thank you for your encouraging comments.

2. In line with the above issue, no data support whether the NEK2 inhibitor is functional inhibition of NEK2 in the in vivo mouse model (Figure 7).

Response (1st Round): We have measured both the expression level and phosphorylation state of NEK2 in tumors that have received combinatorial therapy (including NEK2 inhibitor alone or together with an anti-PD-L1 antibody) in an orthotopic mouse model, and confirmed that NCL 00017509 inhibits the kinase function of NEK2, rather than mediating protein degradation, in vivo.

Reviewer#4's comment: Great. It was addressed experimentally.

Response (2nd Round): Thank you for your encouraging comments.

3. If NEK2 phosphates PD-L1 resulting in its stabilization, the author also need to mutate NEK2 (NEK2-dead: K37R inactive mutant) to repeat and confirm the results shown in the Figure 6.

Response (1st Round): Thank you for your constructive suggestions. As you suggest, we have now used an inactive mutant of NEK2 (K37R, a kinase-dead mutant) in NEK2-depleted KPC cells, and found the mutation indeed caused NEK2 to have loss of function in the regulation of expression, half-life, and ubiquitination of PD-L1, confirming that NEK2 phosphorylates PD-L1, resulting in its stabilization.

Reviewer#4's comment: Great. It was addressed experimentally.

Response (2nd Round): Thank you for your encouraging comments.

4. It is not clear why the author chose to study NEK2 in pancreatic cancer. The rationale is not clear even they showed NEK2 expression is negatively correlated with patient outcome.

Response (1st Round): Our research group focuses on pancreatic diseases, especially pancreatic cancer, in which PD-1/PD-L1 inhibitors show low efficacy and effectiveness. Increasing evidence suggests that post-translational modification of PD-L1 largely determines its targeted immunotherapy. At the beginning of this study, we found that there are two conserved modification motifs of NEK family on PD-L1. Furthermore, using TCGA datasets, we investigated the differential expression and prognostic

relevance of all members of the NEK family in pancreatic cancer, finding potential clinical significance of NEK2 in such disease. In addition, NEK2 participates in the regulation of the cell cycle and mitosis, while regulators of the cell cycle (including but not limited to CDK4/5/6) are closely related to the expression level of PD-L1 and its mediation of cancer immune resistance. Therefore, we chose to study NEK2 in pancreatic cancer immunity. We have now highlighted the rationale of this study in the Introduction and supplemented the whole manuscript with additional information.

Reviewer#4's comment: In the current version, there seems to be no problem with the legitimacy of the study.

Response (2nd Round): Thank you for your encouraging comments.

Reviewer #3 (Remarks to the Author): with expertise in PD-L1 - mechanisms of regulation

The reviewer has three major concerns about this paper:

Response (1st Round): We thank the reviewer for evaluating the manuscript. All the reviewer's concerns have been addressed in a point-by-point manner as follows.

1) The expression of PD-L1 seems to positive in most of the samples shown in the paper, but it is generally believed that PD-L1 is only expressed in a smaller subset of tumors. For instance, the ProteinAtlas database shows that PD-L1 is basically negative in pancreatic cancer. Thus, the correlation between NEK2 and PD-L1 does not seem to be convincing enough.

Response (1st Round): Thank you for your critical comments. Firstly, we disagree that

“The expression of PD-L1 seems positive in most of the samples shown in the paper.”. Actually, we performed a pancreatic cancer tissue microarray (n=156) and found that approximately one half of the cases showed little or no expression of PD-L1, as well as relatively low expression of NEK2, indicating a positive correlation between PD-L1 and NEK2 expression in pancreatic cancer. Furthermore, although the Protein Atlas database shows that PD-L1 is essentially negative in pancreatic cancer, the number of patient samples detected in the Protein Atlas database is rather fewer than in our study using a tissue microarray. In addition, we need to mention that the antibodies used to detect PD-L1 expression in the Protein Atlas and in our study are also different, which may be additional important reason for such a difference. Although both antibodies have been validated as specific against PD-L1, the antibody used in our study has received many citations. We have added the information above throughout the manuscript.

Reviewer#4's comment: The author's answer is reasonable and I have no argument about the positive correlation between PD-L1 and NEK2 in the current version.

Response (2nd Round): Thank you for your encouraging comments.

2) The relationship between the roles of NEK2 and GSK3B should be clarified, plus the experiments showing the relative importance of both regulators on PD-L1 expression.

Response (1st Round): Thank you for your constructive suggestions. As suggested, we have now performed additional experiments to investigate the relative importance of NEK2 and GSK3 β on PD-L1 expression in the context of our study (in other words, in pancreatic cancer). Interestingly, we found that NEK2 inhibition led to inactivation of

GSK3 β (decreased p-GSK3 β) and downregulation of PD-L1 (consistent with our previous results in this study) in a dose-dependent manner. However, the effects of GSK3 β inhibitor on NEK2 and PD-L1 expression are not stable enough for conclusive conclusions to be made. Overall, the results suggest that NEK2, rather than GSK3 β , plays a relatively dominant role in the regulation of PD-L1 expression in pancreatic cancer. We have now supplemented the manuscript with these data.

Reviewer#4's comment: GSK3 β cannot bind to normally glycosylated-PD-L1 and it only phosphorylates T180 and S184 of non-glycosylated-PD-L1. This phosphorylation induces degradation of non-glycosylated PD-L1. This event is restricted to non-glycosylated PD-L1, not glycosylated PD-L1. (Nature Communications. 2016;7:12632. Supplementary Figure5). Logically, GSK3b inhibitor should increase the level of non-glycosylated PD-L1. Additional data showed only the level of glycosylated PD-L1, and this data confuses. I think this data is not essential to support the hypothesis of this manuscript. Alternatively, if phosphorylation by NEK2 can affect the glycosylation of PD-L1, it is advisable to check the binding of GSK3b to PD-L1.

Response (2nd Round): We are very appreciated with this important suggestion and agree with it. We have now checked the binding of GSK3b to PD-L1 with or without NEK2 inhibition, and co-IP assay showed that NEK2 inhibitor indeed affects the PD-L1-GSK3 β interaction (**Supplementary Fig. 9c, d**). Given the potential direct interplay between NEK2 and GSK3 β , as well as the different cancer-specific contexts, to address their importance on PD-L1 regulation may be over-qualified in our study.

3) NEK2 has been established as an anti-cancer target, and it is unclear if NEK2 may regulate anti-tumor immunity through any other signaling pathways. For example,

previous studies have implicated Nek2 in early B cell development and germinal center formation. <https://www.ncbi.nlm.nih.gov/pmc/articles/PMC4251609>

Response (1st Round): Thank you for your insightful comments. We have now rechecked the previous data for NEK2 overexpression-induced differential expression of proteins, in which we identified PD-L1 as a potential target of NEK2 in pancreatic cancer. Gene Ontology enrichment analysis revealed that NEK2 is closely related to multiple aspects of immune regulation, including but not limited to, cytokine and cytokine receptor binding, interleukin production, leukocyte proliferation, migration, and activation, in addition to phagocytosis and endocytosis. We also need to mention that NEK2 expression has also been observed in suppressive immune cells (such as macrophages, DCs, and MDSCs) in our study, suggesting that NEK2 may be involved in tumor immune resistance through regulation of these immunosuppressive populations. We have now supplemented the manuscript with these aspects.

Reviewer#4's comment: I think authors do not need to be defensive about this comment. Of course, NEK2 is probably multifunctional and NEK2i could affect other parts related to the anti-tumor effect. Reviewer #3 may want to discuss this point in the extended view beyond the PD-L1 regulation mechanism. Such an extended discussion would give deep insight into this manuscript.

Response (2nd Round): We deeply thank you for your insightful comments and critical suggestion. An extended discussion of NEK2 beyond PD-L1 regulation has been supplemented in discussion section as you suggested, as following: As mentioned above, NEK2 is a multifunctional protein involved in non-immune and immune function including cell cycle regulation, microtubule stabilization, kinetochore attachment, spindle assembly checkpoint, phosphorylation of downstream proteins and

the maintenance of normal development and function of B cells, and has been associated with tumor progression and clinical prognosis in multiple cancers.

2. Reviewers #4's comments.

General comment

In this manuscript, authors identify the new regulatory mechanisms of PD-L1 in PDAC, and based on this finding, they suggest a potential combination to enhance the efficacy of ICB against PDAC. There seems to be no problem with the legitimacy of the study and the correlation between NEK2 and PD-L1 is clearly shown through multiple approaches including database, TMA, animal model, and in vitro assay. The effect of NEK2 on PD-L1 and anti-tumor immunity is recognized. However, there is a fundamental problem with their hypothesis, which can be controversial. To resolve this issue, I think we need additional experimental evidence.

Response (2nd Round): Thank you for your encouraging and critical comments.

Major comment

Because T194/T210 is located in the extracellular domain of PD-L1, T194/T210 is only exposed inside of ER and Golgi lumen during transport into the plasma membrane. It is probably rare that T194/T210 is exposed to the cytosolic part. Therefore, Author should show evidence NEK2 can directly bind to PD-L1 and phosphorylates T194/T210 of PD-L1 in ER and/or Golgi.

Response (2nd Round): Thank you for your critical comments. As requested, all your concerns have been addressed in a point-by-point manner as follows.

1. Authors need to check the localization of NEK2 in ER or/and Golgi. Trypsinization

assay with ER fraction will be available to check this issue. The Journal of Clinical Investigation. 2019 Jul 15;129(8):3324-3338, Molecular Cell. 2018;71(4):606-620

Response (2nd Round): We deeply thank you for your attentive review and constructive suggestion. Trypsinization assay with ER fraction have been performed as your suggested. The results are as follows: no signals were detected with antibodies against both cytosolic and luminal proteins after trypsinization in the permeable fraction. In contrast, signals for the cytosolic domain of IRE1 α were rapidly reduced after trypsinization, whereas signals for NEK2 and HSP90B1 were maintained in the non-permeable fraction (**Supplementary Fig. 8a**). Therefore, NEK2 directly interact with PD-L1 inside the ER lumen in pancreatic cancer.

2. Although they show the binding between PD-L1 and NEK2 with endo-IP, actually, this analysis cannot exclude the involvement of other factors. Traditional GST (or His)-full down assay with purified PD-L1 and NEK2 will be suitable and Duo-link assay with antibodies against PD-L1 and NEK2 will support real binding in the cell, not artificial tube.

Response (2nd Round): We deeply thank you for your insightful suggestion. For further supporting our results, GST-pull down assay and Duo-link assay have been performed as you suggested, confirming the direct interaction and colocalization of NEK2 and PD-L1 without the involvement of other factors (**Fig. 4h, 4i**).

3. All current data related to phosphorylation is indirect evidence to show NEK2 signaling can phosphorylate PD-L1. Other factors in NEK2 signaling may phosphorylate T194/T210. Authors have already acquired specific Abs against T194/T210 and active NEK2 (14-545M, sigma MERK) and PD-L1 are commercially

available, and therefore, why not perform *in vitro* kinase assay. This result will be critical evidence to clearly show NEK2 directly phosphorylate T194/T210 of PD-L1.

Response (2nd Round): We deeply thank you for meticulous review and in-depth suggestion. The *in vitro* kinase assay for NEK2 and PD-L1 has been supplemented as your suggested. After addition of NEK2, the phenomenon of PD-L1 phosphorylation has been observed (**Fig. 6d**). Therefore, this result show NEK2 directly phosphorylates PD-L1.

4. If authors acquire these data, it is better to gain all data related to binding and phosphorylation and reorganize them in another figure.

Response (2nd Round): We deeply thank you for your critical suggestion. All data related to binding and phosphorylation have been supplemented and reorganized in **Fig. 6**.

I think this issue had to be indicated in the first revision. However, with these evidences, it will be a complete manuscript and there will be no boring argument related to phosphorylation of an extracellular domain in the respect of the transmembrane protein-topology.

Response (2nd Round): Thank you for your encouraging comments. All your concerns have now been addressed in a point-by-point manner.

Minor comment

In the main text, there is no description of Supplementary Fig.8.

Response (2nd Round): We deeply thank you for your careful review and sincerely apologize for our negligence. The description of primary Supplementary Fig.8 has now

been supplemented at the position of **Supplementary Fig.10** in update version.

REVIEWERS' COMMENTS

Reviewer #4 (Remarks to the Author):

In the last version, the major issues were how NEK2 can bind with T194/T210 in the ER lumen and evidence was not enough to conclude direct phosphorylation of T194/T210 by NEK.

To address these critical issues, in this version, the authors showed the localization of NEK in the ER lumen through the trypsinization assay with ER fraction, showing that NEK2 can interact with T194/T210 of PD-L1 inside the ER lumen (Fig S8a). In addition, the GST-pulldown assay and Duo-Link assay (Fig. 4h and i) support that NEK2 can directly bind with PD-L1. Moreover, the in vitro kinase assay nicely showed that NEK2 can directly phosphorylate T194/T210 of PD-L1 (Fig.6d). Collectively, we can conclude that NEK2 directly binds to that extracellular domain of PD-L1 in the ER lumen and phosphorylate T194/T210 of PD-L1.

The authors answer my picky questions with clear experimental evidence. I believe this revised manuscript has been greatly improved with this important evidence. I sincerely appreciate to author's faithful attitude and scientific passion.

REVIEWER COMMENTS AND POINT-BY-POINT RESPONSE

Reviewer #4 (Remarks to the Author):

In the last version, the major issues were how NEK2 can bind with T194/T210 in the ER lumen and evidence was not enough to conclude direct phosphorylation of T194/T210 by NEK.

To address these critical issues, in this version, the authors showed the localization of NEK in the ER lumen through the trypsinization assay with ER fraction, showing that NEK2 can interact with T194/T210 of PD-L1 inside the ER lumen (Fig S8a). In addition, the GST-pulldown assay and Duo-Link assay (Fig. 4h and i) support that NEK2 can directly bind with PD-L1. Moreover, the in vitro kinase assay nicely showed that NEK2 can directly phosphorylate T194/T210 of PD-L1 (Fig.6d). Collectively, we can conclude that NEK2 directly binds to that extracellular domain of PD-L1 in the ER lumen and phosphorylate T194/T210 of PD-L1.

The authors answer my picky questions with clear experimental evidence. I believe this revised manuscript has been greatly improved with this important evidence. I sincerely appreciate to author's faithful attitude and scientific passion.

Response (3rd Round): We deeply thank you for your inspiring comments, constructive suggestions, and everything you have done during the review process for improving our manuscript.